# Deep brain stimulation-guided optogenetic rescue of parkinsonian symptoms

Sébastien Valverde[1,4], Marie Vandecasteele[1,4], Charlotte Piette[1,2,4], Willy Derousseaux[1], Giuseppe Gangarossa[1], Asier Aristieta Arbelaiz[1], Jonathan Touboul[2,5], Bertrand Degos[1,3,5] & Laurent Venance[1,5 ✉]

Deep brain stimulation (DBS) of the subthalamic nucleus is a symptomatic treatment of Parkinson's disease but benefits only to a minority of patients due to stringent eligibility criteria. To investigate new targets for less invasive therapies, we aimed at elucidating key mechanisms supporting deep brain stimulation efficiency. Here, using in vivo electrophysiology, optogenetics, behavioral tasks and mathematical modeling, we found that subthalamic stimulation normalizes pathological hyperactivity of motor cortex pyramidal cells, while concurrently activating somatostatin and inhibiting parvalbumin interneurons. In vivo opto-activation of cortical somatostatin interneurons alleviates motor symptoms in a parkinsonian mouse model. A computational model highlights that a decrease in pyramidal neuron activity induced by DBS or by a stimulation of cortical somatostatin interneurons can restore information processing capabilities. Overall, these results demonstrate that activation of cortical somatostatin interneurons may constitute a less invasive alternative than subthalamic stimulation.

[1] Dynamics and Pathophysiology of Neuronal Networks Team, Center for Interdisciplinary Research in Biology, Collège de France, CNRS UMR7241/INSERM U1050, MemoLife Labex, 75005 Paris, France. [2] Department of Mathematics and Volen National Center for Complex Systems, Brandeis University, Waltham, MA, USA. [3] Department of Neurology, Avicenne University Hospital, Sorbonne Paris Nord University, 93009 Bobigny, France. [4]These authors contributed equally: Sébastien Valverde, Marie Vandecasteele, Charlotte Piette. [5]These authors jointly supervised this work: Jonathan Touboul, Bertrand Degos, Laurent Venance. ✉email: laurent.venance@college-de-france.fr

Parkinson's disease results from the neurodegeneration of the nigro-striatal dopaminergic neurons. The main symptomatic treatment for Parkinson's disease consists in substituting lacking dopamine with levodopa and/or dopaminergic agonists, but after a typical "honeymoon" period with dopaminergic therapy, patients inevitably develop motor complications[1]. At this stage, deep brain stimulation at high frequency of the subthalamic nucleus (DBS) constitutes to date the most efficient symptomatic treatment[2,3]. However, due to its surgical invasiveness and strict eligibility criteria, DBS benefits only to a minority of patients (~5–10%). Several hypotheses have been proposed to explain the beneficial effects of DBS[4–6]. Notably, a growing body of evidence points towards a cortical effect of DBS in both parkinsonian rodent models[7–11] and patients[12–17]. Here, we reasoned that mimicking the cortical effects of DBS should reproduce its therapeutic benefits, thus paving the way for less invasive approaches.

For this purpose, we (i) determined DBS effects on cortical cell-type specific populations using a combination of in vivo electrophysiological and optogenetic approaches, (ii) reproduced these effects using optogenetics in freely-moving parkinsonian mice, and (iii) explored mathematically how DBS and DBS-guided optogenetics could restore cortical information processing capabilities. We showed that DBS normalized pathological hyperactivity of motor cortex pyramidal cells, while concurrently inhibiting parvalbumin (PV)- and activating somatostatin (SST)-expressing GABAergic interneurons. Furthermore, reproducing these effects by direct opto-activation of cortical SST interneurons alleviates motor symptoms in a parkinsonian mouse model. Lastly, our computational model shows that the dampening of the firing activity of pyramidal cells by DBS and DBS-guided optogenetics restores cortical information processing capabilities. Overall, these results establish that cortical SST interneurons constitute a promising target for a less invasive alternative to DBS.

## Results

**DBS decreases pathological hyperactivity of pyramidal cells**. To understand how DBS affects cortical activity, we first aimed at depicting the electrophysiological signature of Parkinson's disease at the neuronal level in the primary motor cortex (M1). We performed in vivo single unit juxtacellular recordings of pyramidal neurons in deep layers of M1 in a rat model of Parkinson's disease (Fig. 1a and Supplementary Fig. 1a) by unilateral stereotaxic injection of 6-hydroxydopamine (6-OHDA) in the substantia nigra pars compacta (SNc). In anesthetized 6-OHDA-lesioned rats, we observed an increase in the spontaneous firing rate of M1 pyramidal cells ($n = 41$) compared to sham animals ($n = 36$) ($p = 0.0014$) (Fig. 1b), in line with previous observations[10,18]. We then explored the effect of DBS (STN stimulation parameters: 2–4 V, 60 μs at 130 Hz during 2 min) on this pathophysiological hyperactivity of M1 pyramidal neurons. In 6-OHDA-lesioned rats, we found that the increased firing activity was diminished by DBS ($p = 0.0311$, $n = 20$) back to physiological firing rates (Fig. 1b), with 68% of pyramidal cells inhibited by DBS (Fig. 1c). DBS also decreased M1 neuron firing rate in sham animals ($p = 0.0266$, $n = 19$), and the proportion of inhibited neurons, as well as the change in firing rate, were similar in sham and 6-OHDA-lesioned rats ($p = 1$ and $p = 0.7894$, respectively) (Supplementary Fig. 1b, c). Overall, DBS diminished the firing rate of pyramidal cells in sham rats and normalized their pathophysiological hyperactivity in parkinsonian rats.

**GABAergic circuits mediate DBS inhibition of pyramidal cells**. We next investigated the mechanistic underpinnings related to the decreased activity of pyramidal cells under DBS. For this purpose, we performed single-cell in vivo intracellular recordings of electrophysiologically identified M1 pyramidal cells in anesthetized rats (Fig. 2a). Spontaneous firing of pyramidal cells was determined before and during 150 s of DBS. We further confirmed that DBS decreased the spontaneous firing rate ($p = 0.0002$, $n = 20$) of M1 pyramidal neurons (Fig. 2b). This decrease was accompanied by a hyperpolarization of $-3.7 \pm 1.0$ mV of their membrane potential ($p = 0.0007$, $n = 20$) (Fig. 2b), and a decrease in their membrane time constant ($p = 0.0390$, $n = 18$), input resistance ($p = 0.0497$, $n = 19$) and AP threshold ($p = 0.0263$, $n = 20$), without affecting their Ih current ($p = 0.2336$, $n = 17$). In a subset of cells ($n = 3$) that were antidromically activated by STN stimulation, the antidromically-evoked action potentials were rapidly shunted and a marked hyperpolarization was observed (Supplementary Fig. 1d). We further characterized DBS-induced changes in pyramidal cell excitability by applying successive depolarizing current steps before and during continuous DBS (Fig. 2c). DBS induced a decrease in depolarization-evoked activity, associated with an increased rheobase ($p = 0.0312$, $n = 17$), without affecting the gain ($p = 0.1746$, $n = 17$) of pyramidal cell f–I curve. Overall, the decrease of several properties of pyramidal cell excitability under DBS could participate in the decrease of their firing activity.

To determine the evoked conductances in M1 pyramidal cells, we next applied single STN stimulations (2–4 V, 60 μs) (Fig. 2d, e). We observed a post-stimulation hyperpolarization, sufficient to delay the evoked firing activity ($p = 0.0057$, $n = 6$) (Fig. 2d). Analysis of the voltage-dependency showed that the early phase of the evoked postsynaptic responses reversed at $-71.7 \pm 2.7$ mV ($n = 9$) (Fig. 2e), which corresponds to the chloride reversal potential[19]. These results suggest that DBS recruits GABAergic circuits responsible for the hyperpolarization of pyramidal cells and the increase in rheobase, leading to the reduced activity observed in both in vivo juxtacellular and intracellular recordings of M1 pyramidal cells.

**Somatostatin interneurons are activated by DBS**. In M1, GABAergic inhibition is provided by local neuronal populations mainly composed of PV and SST interneurons[20]. To identify the cell-type specific populations recruited by DBS, we used genetically modified mice expressing channelrhodopsin (ChR2) in either PV (*Pv::ChR2* mice) or SST (*Sst::ChR2* mice) interneurons (Fig. 3, Supplementary Figs. 2 and 3). We ensured that ChR2 was expressed in the targeted populations with minimal non-specific expression (Supplementary Fig. 2).

In mice, stereotaxic targeting of the STN for DBS is challenging. To ensure the proper placement of the stimulation electrode, we first lowered a microelectrode to electrophysiologically localize the STN based on typical STN neuron firing (Supplementary Fig. 3a), and in a second step implanted the DBS electrode at the same coordinates (Supplementary Fig. 3b). We then performed in vivo juxtacellular recordings of M1 neurons in anesthetized *Pv::ChR2* and *Sst::ChR2* mice to monitor DBS-evoked responses in opto-identified neuronal subpopulations (Figs. 3a, b and Supplementary 3c, d). Namely, once the DBS electrode was inserted in STN, an optical fiber placed on top of M1 shone light (100 ms at 0.5 Hz) and opto-responsive neurons were detected by a recording microelectrode lowered within M1. PV and SST interneurons were distinguished from pyramidal neurons by their responses to light in *Pv::ChR2* and *Sst::ChR2* mice, and by post-hoc clustering of their waveform characteristics based on principal component analysis (Supplementary Fig. 3c, d). A subset of PV, SST, and pyramidal neurons were juxtacellularly labeled with neurobiotin for immunohistochemical

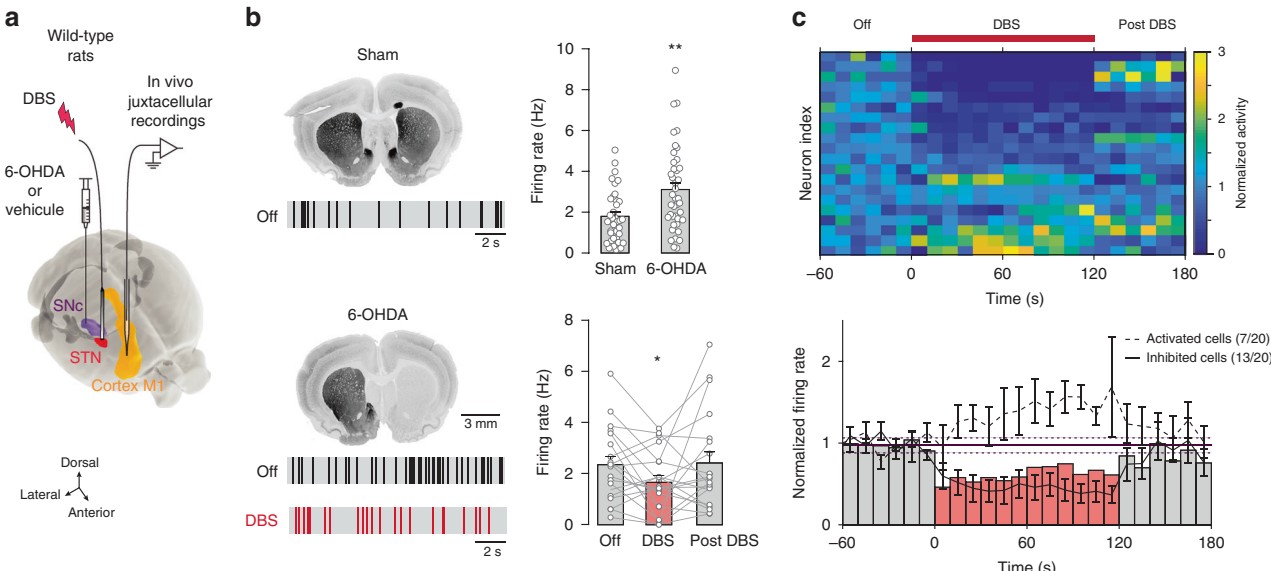

**Fig. 1 DBS mediates in vivo inhibition M1 pyramidal cells in rats. a** In vivo experimental set-up in anesthetized adult rats. **b** TH immunostaining and representative raster plots (15 s) of M1 neurons activity recorded in sham rats and in 6-OHDA-lesioned rats with or without DBS. Spontaneous activity of cortical neurons (mean ± SEM and individual neurons are represented) was higher in 6-OHDA-lesioned ($n = 41$ neurons) compared to sham ($n = 36$) rats ($p = 0.0014$, $t$-test); DBS decreased the hyperactivity observed in 6-OHDA-lesioned rats (Off vs DBS: $p = 0.0311$, DBS vs Post: $p = 0.0811$, paired $t$-test, $n = 20$ neurons). **c** Heatmap of individual cortical neurons normalized firing rate (top), and averaged time course (bottom), before, during and after DBS (10s bins) in 6-OHDA-lesioned rat: bars correspond to the median of all neurons (mean ± 2×SD of the baseline is indicated) and separate time courses (mean ± SEM) for activated and inhibited neurons are superimposed. Under DBS, 65% of the pyramidal cells were inhibited while 35% were activated ($n = 20$). All statistical tests are two-tailed. *$p < 0.05$; **$p < 0.005$.

and morphological identification and used as ground truths for the principal component analysis (Fig. 3c and Supplementary Fig. 3c–e). Most (75%) of the recorded neurons were located in M1 layer V ($n = 72$ neurons subjected to DBS; Supplementary Fig. 3e). In agreement with in vivo recordings in anaesthetized mice[21], SST interneurons exhibited a lower spontaneous activity compared to PV cells ($p = 0.0002$, PV and SST median firing rates were 1.23 and 0.05 Hz, respectively; Fig. 3d). Consistent with our observations in rats, DBS (60 μs, 120 μA at 130 Hz) decreased the firing activity of pyramidal cells in the transgenic mouse lines ($p = 0.0297$, $n = 28$; Fig. 3d–f). We then investigated DBS effect on PV interneurons. Surprisingly, DBS decreased their activity ($p = 0.0074$, $n = 28$; Fig. 3d–f). We next examined SST interneurons and observed that DBS caused the opposite effect, i.e. increasing SST firing ($p = 0.0040$, $n = 26$; Fig. 3d–f). These effects were robust across cell populations as 64% of the SST cells were excited and 70% of the PV cells were inhibited (Fig. 3f). The activity of pyramidal, PV, and SST cells was stable under DBS, and reversible (with a shorter delay after DBS offset for PV cells than for SST and pyramidal neurons). Interestingly, the kinetics of DBS effects on pyramidal cells mirrored those of SST interneurons: we observed similar delays between DBS onset and its effects on pyramidal and SST cells, and also at DBS offset (Fig. 3e). Consistently, the modulation of pyramidal cells was strongly correlated with SST interneuron activity during and after DBS ($r = -0.91$, $p = 1.0 \times 10^{-9}$). Therefore, DBS efficiently drives activation of M1 SST interneurons, concurrently inhibiting M1 pyramidal cells.

**Activity of PV and SST cells in sham and parkinsonian mice.** Since DBS both normalizes pyramidal cell hyperactivity and differentially recruits PV and SST interneurons, we further explored the firing rate modulation of interneurons in parkinsonian conditions. Indeed, hyperactivity of pyramidal cells observed in 6-OHDA-lesioned animals could be caused by a

decreased activity of GABAergic interneurons. We performed in vivo juxtacellular recordings of neurons opto-identified and clustered by principal component analysis as PV or SST cells in anesthetized sham (PV cells = 27 and SST cells = 26) and 6-OHDA-lesioned (PV cells = 14 and SST cells = 15) *Pv::ChR2* and *Sst::ChR2* mice (Fig. 4a, b). The spontaneous firing activity of PV cells was similar in sham and parkinsonian *Pv::ChR2* mice ($p = 0.8391$ with $n = 32$ in sham and $n = 14$ in 6-OHDA-lesioned mice). Similarly, the spontaneous firing rate of SST cells was not different in sham and 6-OHDA-lesioned mice ($p = 0.2112$ with $n = 32$ in sham and $n = 15$ in 6-OHDA-lesioned mice). Pyramidal hyperactivity in parkinsonian models is not likely due to changes in electrophysiological activity of cortical GABAergic populations.

**Inhibitory evoked-responses by PV or SST opto-activation.** To investigate the synaptic weight of the inhibitory inputs onto pyramidal cells, we performed in vivo patch-clamp whole-cell recordings in anesthetized *Pv::ChR2* and *Sst::ChR2* mice (Fig. 4c–f). We recorded pyramidal cells upon opto-activation of PV or SST interneurons. First, we characterized the evoked-PSP following single opto-stimulation (3–20 ms). The opto-activation of PV and SST cells evoked PSPs of similar amplitudes in pyramidal cells, regardless of the membrane potential ($p = 0.3397$, $n = 8$ pyramidal cells in *Pv::ChR2* vs. 6 in *Sst::ChR2* held at $-95/-80$ mV; $p = 0.8632$, $n = 7$ vs. 7 held at $-65/-50$ mV and $p = 0.8838$, $n = 7$ vs. 9 held at $-40/-30$ mV) (Fig. 4c). We analyzed the voltage-dependency of opto-PSPs, which reversed at $-71.5$ mV in *Pv::ChR2* ($n = 6$ pyramidal cells) and $-75.2$ mV in *Sst::ChR2* ($n = 4$) mice ($p = 0.0542$) (Fig. 4c), close to the calculated chloride reversal, $-71.9$ mV. Single-pulse opto-activation of PV ($n = 7$) and SST ($n = 6$) cells induced PSPs whose area, amplitude, rise time and delay to peak increased with increasing opto-pulse duration (2-way repeated-measures ANOVA; Fig. 4d). There was no difference between PSPs evoked by PV or SST

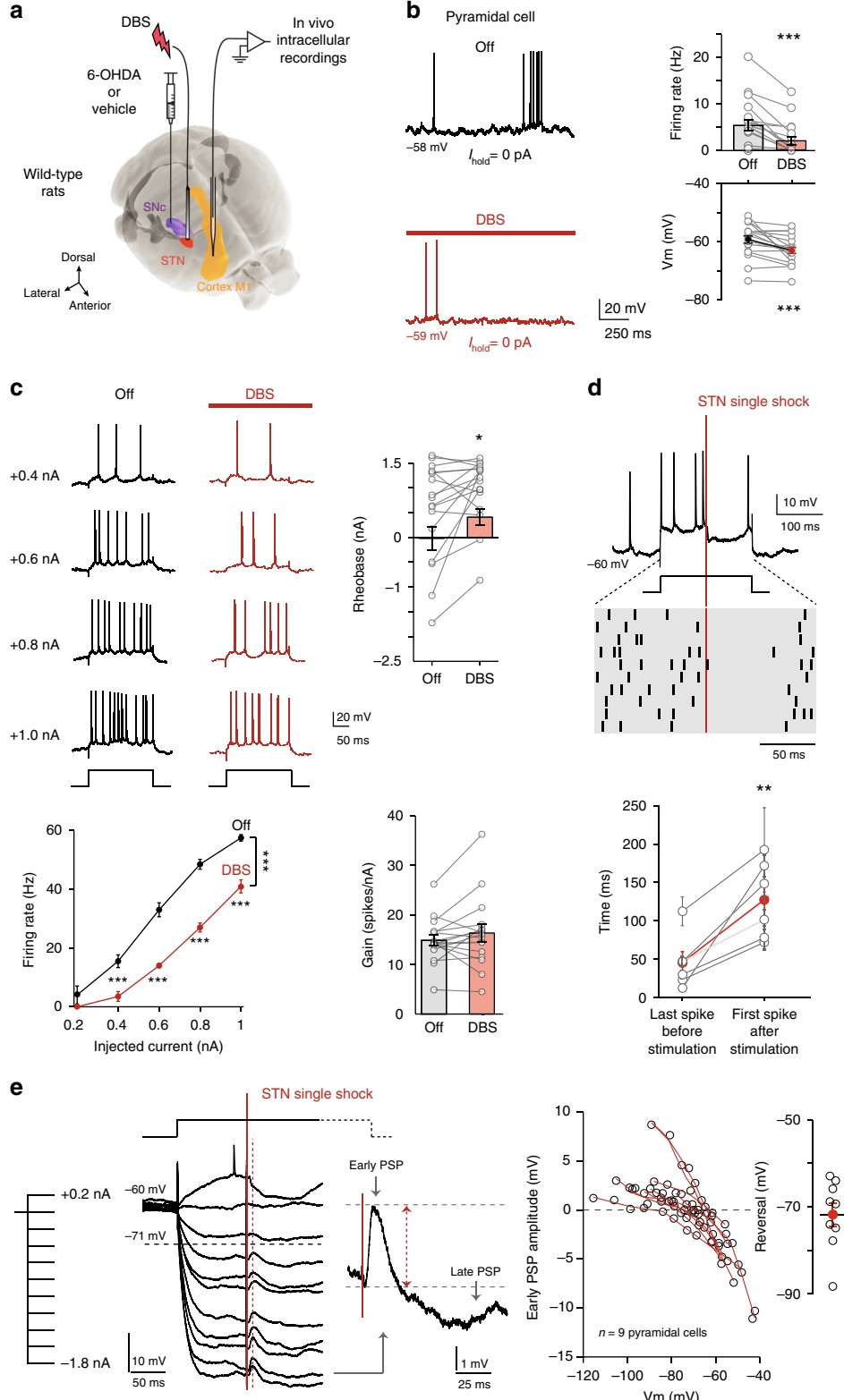

opto-activation, except for the delay to peak amplitude which was shorter upon PV than SST opto-activation ($F_{1,11} = 66.39$, $p < 0.0001$). We then investigated the effects of opto-activation of PV or SST cells at 67 Hz for 5 s (see "Methods" and Supplementary Fig. 4a–d for opto-stimulation frequency selection) (Fig. 4e–i). We observed that opto-activation of PV or SST cells induced a hyperpolarization of similar peak amplitude ($p = 0.2270$, Fig. 4f), with a shorter delay upon PV activation ($p = 0.0124$; Fig. 4g) that

strongly inhibited the firing activity of pyramidal cells ($p = 0.0392$, $n = 5$ in *Pv::ChR2* mice, $p = 0.001$, $n = 8$ in *Sst::ChR2* mice) (Fig. 4e). However, in the later part of the pulse, opto-activation of SST cells induced a less pronounced membrane hyperpolarization than opto-activation of PV cells ($p = 0.0371$; Fig. 4h), together with a partial release of the spiking inhibition only in *Sst::ChR2* mice ($p = 0.0326$; Fig. 4e). In addition, the variance of the membrane potential was larger under opto-

**Fig. 2 Intracellular mechanisms of DBS inhibition of M1 pyramidal cells in rats. a** In vivo experimental set-up. **b** M1 pyramidal neurons recorded intracellularly display a decreased spontaneous activity (Wilcoxon signed-rank test, $n = 20$, $p = 0.0002$), and hyperpolarized membrane potential ($p = 0.0007$) during DBS (mean ± SEM and individual neurons); 95% were spontaneously active. **c** Pyramidal cell transfer function quantified with and without DBS. Depolarizing current steps (200 ms, 0–1.2 nA in 0.2 nA steps) were applied at 1 Hz before and during 100 s of DBS. DBS decreases evoked spiking activity. Left: raw data example and f–I relationship in a pyramidal neuron, showing a decreased evoked firing rate during DBS (DBS effect $F_{1,90} = 155.72$, $p = 2.5 \times 10^{-21}$, 2-way ANOVA followed by Bonferroni-corrected post-hoc tests: all $p < 10^{-3}$ for 0.4–1 nA injected current). Right: DBS increases the rheobase ($p = 0.0312$, Wilcoxon signed-rank test, $n = 17$ neurons), without affecting the gain of the f–I curve ($p = 0.1746$, paired t-test) (mean ± SEM and individual neurons). Rheobase and gain were respectively determined as the x-intercept and the slope of the f-I curve after a linear fit. **d** Single-shock STN stimulation induces a pause in M1 pyramidal evoked firing. Top: representative trial and raster plot of current-evoked activity for 10 trials in the same neuron. Bottom: the interval between the STN stimulation and the next evoked-spike is longer than between the STN stimulation and the previous evoked spike ($p = 0.0057$, paired t-test, $n = 6$ neurons) (mean ± SEM and individual neurons). **e** Voltage-dependency of the potential evoked by single-shock STN stimulation. Left: a single-shock STN stimulation (red line) applied during hyperpolarizing or depolarizing current steps (−1.8 to +1.0 nA in 0.2 nA increments, mean of 10 trials) elicits an evoked potential (dashed line). Zoom of the PSP evoked by STN single stimulation for −1.6 nA injected current. The amplitude of the early phase of the PSP is inversely correlated with the membrane potential when the stimulation was applied (voltage-dependency of 9 individual neurons). Right: the STN-evoked potential reverses around −71.7 mV ($n = 9$ neurons) (mean ± SEM and individual neurons). DBS artifacts were removed in **b**, **c** for clarity. All statistical tests are two-tailed. *$p < 0.05$; **$p < 0.005$; ***$p < 0.001$.

activation of SST than PV cells ($p = 0.0371$; Fig. 4i), denoting a strong shunting of synaptic activity reaching the soma by PV opto-activation, whereas some synaptic activity subsisted (leading eventually to spikes) under SST opto-activation.

**M1 SST cell opto-activation alleviates parkinsonian symptoms.** Since DBS activates SST interneurons, we hypothesized that cortical SST interneurons could constitute a target to efficiently mimic the effects of DBS. Therefore, we tested whether the direct activation of GABAergic interneurons would improve motor symptoms in freely moving parkinsonian mice. We first compared the respective effects of SST and PV opto-activation at 67 Hz (see "Methods" and Supplementary Fig. 4a–d for opto-stimulation frequency selection). For this purpose, unilaterally 6-OHDA-lesioned mice (*Sst::ChR2*, *Pv::ChR2* and wild-type mice) were ipsilaterally implanted with an optical fiber in M1 (Fig. 5a). We monitored the impact of opto-stimulation on the asymmetrical locomotor behavior induced by unilateral 6-OHDA-lesioning in three different tasks: the open field (Fig. 5b–g), cylinder test (Fig. 5h–m) and cross-maze (Fig. 5n–s).

Opto-activation of SST interneurons in 6-OHDA-lesioned *Sst::ChR2* mice successfully decreased their asymmetrical locomotor behavior. Indeed, in the open field, opto-activation of SST cells in 6-OHDA-lesioned *Sst::ChR2* mice decreased spontaneous ipsilateral rotations ($n = 11$ mice; rotations/min: $p = 0.001$; Fig. 5c and normalized rotations: $p = 0.0002$; Supplementary Fig. 4e). This decrease in asymmetrical behavior did not result from a decreased locomotor activity, which remained unaffected by opto-activation of SST cells ($p = 0.5721$; Fig. 5c). In the cylinder test, the opto-activation of SST cells decreased the asymmetry in front paw usage preference ($p = 0.0156$, $n = 9$; Fig. 5i). In the cross-maze, 6-OHDA-lesioned *Sst::ChR2* mice ($n = 11$) exhibited a strong bias towards ipsilateral turns (Fig. 5o), as expected for hemi-parkinsonian rodents, while sham-lesioned *Sst::ChR2* mice ($n = 5$) displayed no turn preference (Supplementary Fig. 4g). During SST opto-activation, a decrease in the ipsilateral bias ($p = 0.0019$) and an increase in the straight choice ($p = 0.0022$; Fig. 5o), were observed.

PV interneuron opto-activation only partially reproduced the improvement in motor symptoms induced by SST opto-activation. In 6-OHDA-lesioned *Pv::ChR2* mice, opto-activation decreased spontaneous ipsilateral rotations ($n = 12$; rotations/min: $p = 0.0123$; Fig. 5d and normalized rotations: $p = 0.0125$; Supplementary Fig. 4e) without affecting locomotor activity ($p = 0.0737$; Fig. 5d). However, opto-activation of PV cells did not

decrease the asymmetrical locomotor behavior in the cylinder test ($p = 0.9139$, $n = 9$; Fig. 5j) or in the cross-maze task ($n = 12$; $p = 0.1876$ for ipsilateral, $p = 0.3419$ for straight, and $p = 0.0960$ for contralateral turn preference; Fig. 5p).

Interestingly, opto-activation of cortical interneurons in *Sst::ChR2* or *Pv::ChR2* sham-mice did not induce an asymmetrical behavior contralateral to the opto-activation neither in the open field (Supplementary Fig. 4f) nor in the cross-maze task (Supplementary Fig. 4g). This indicates that the reduced asymmetry observed upon SST opto-activation in 6-OHDA-lesioned mice is due to an improvement of pathological symptoms rather than a generic effect of unilateral opto-activation of cortical interneurons.

We also ensured that opto-stimulation in wild-type (non-opsin expressing) 6-OHDA-lesioned mice, did not affect spontaneous ipsilateral rotations ($n = 11$; rotations/min: $p = 0.5591$; Fig. 5e and normalized rotations: $p = 0.2540$; Supplementary Fig. 4e) or locomotor activity ($p = 0.6289$; Fig. 5e) in the open field, asymmetry in front paw usage preference in the cylinder test ($p = 0.3841$, $n = 9$; Fig. 5k) and asymmetrical locomotor behavior in the cross-maze ($n = 11$; $p = 0.8292$ for ipsilateral, $p = 0.9755$ for straight, and $p = 0.3409$ for contralateral turn preference; Fig. 5q).

We next evaluated the effects of DBS and levodopa treatments in wild-type 6-OHDA-lesioned mice in open-field, cylinder and cross-maze tasks, for comparison with opto-activation of SST and PV cells. In the open-field, DBS did not reduce the spontaneous ipsilateral rotations ($n = 12$; rotations/min: $p = 0.2082$; Fig. 5f and normalized rotations: $p = 0.2456$; Supplementary Fig. 4e) but increased the locomotor activity ($p = 0.0438$; Fig. 5f), in line with a recent report[22]. DBS did not modify asymmetrical locomotor behavior for the cylinder test ($n = 12$, $p = 0.9768$; Fig. 5l) or in the cross-maze ($n = 12$; $p = 0.2040$ for ipsilateral, $p = 0.7675$ for straight, and $p = 0.3388$ for contralateral turn preference; Fig. 5r). Levodopa treatment (6 mg/kg) in 6-OHDA-lesioned mice efficiently reversed the spontaneous ipsilateral rotations into contralateral rotations ($n = 10$; rotations/min: $p = 0.0004$; Fig. 5g and normalized rotations: $p = 2.0 \times 10^{-5}$; Supplementary Fig. 4e) and increased the locomotor activity ($p = 0.0135$) in the open-field. The asymmetry in front paw usage preference was reversed to contralateral preference ($n = 10$, $p = 0.0002$; Fig. 5m) in the cylinder test, as well as the ipsilateral bias ($n = 7$, $p = 0.0029$) in favor of contralateral ($p = 0.0029$) turn preference in the cross-maze (Fig. 5s).

Therefore, specific opto-activation of cortical SST interneurons alleviates the asymmetrical behavior while electrical DBS increases locomotor activity in parkinsonian mice.

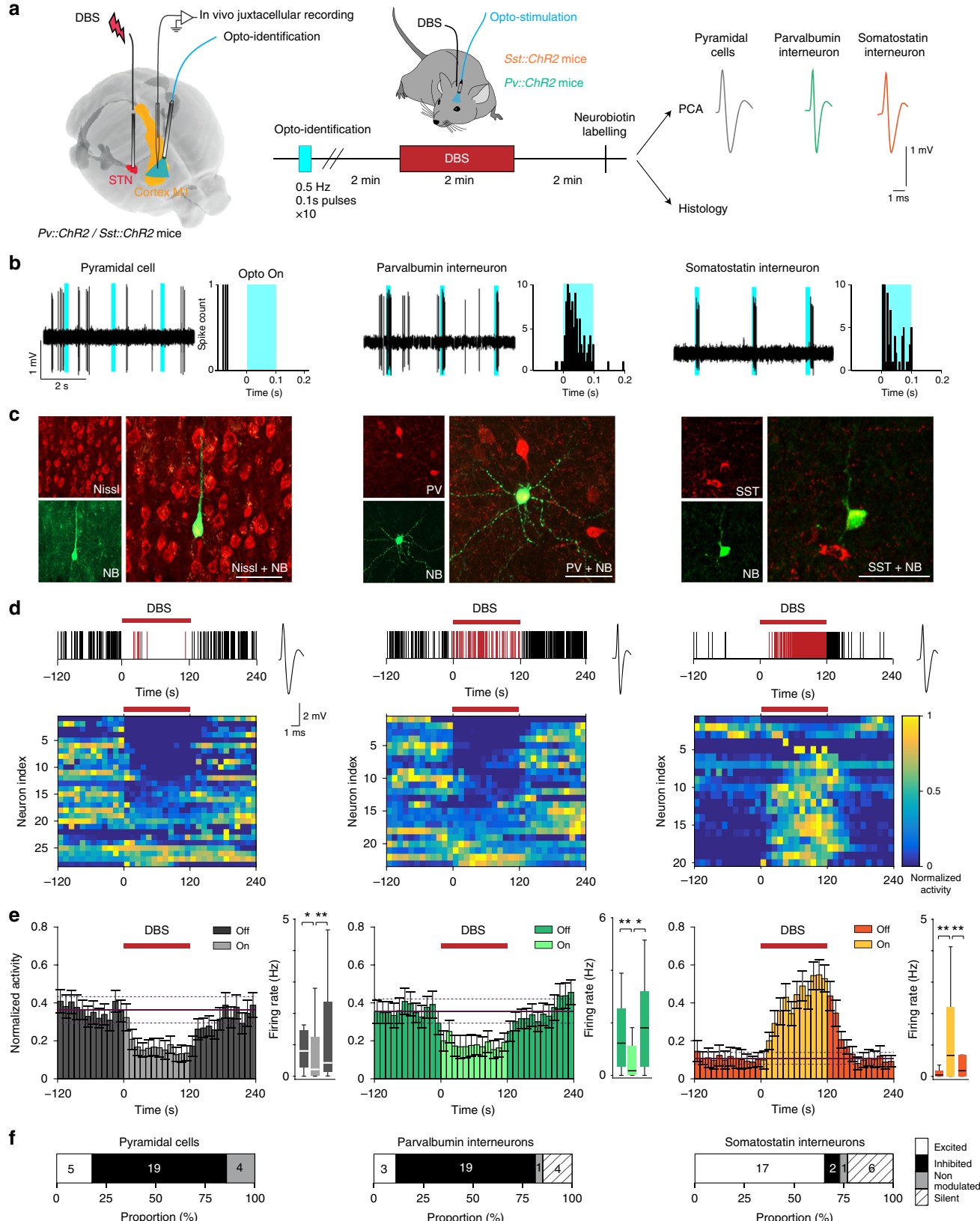

**M1 L5 model recapitulates DBS effects on cortical firing**. To better understand the beneficial effects of DBS and SST opto-activation, we theoretically tested the information processing capabilities of cortical networks in control and parkinsonian conditions under DBS and when activating PV or SST

interneurons. To this purpose, we built a simplified spiking neural network model of L5 motor cortex, including pyramidal, PV, and SST cells modeled as adaptive exponential integrate-and-fire neurons (Fig. 6a). The choices of intrinsic parameters defining each population (Supplementary Table 1), network architecture

**Fig. 3 In vivo DBS activates somatostatin interneurons in mice. a** In vivo experimental setup. A bipolar electrode is lowered into the STN and an optical fiber is placed over M1, while recording from neurons in M1. **b**, Top: electrophysiological traces of representative opto-identified pyramidal, SST, and PV neurons recorded in M1. SST and PV neurons are opto-activated by brief flashes of light (shown in blue). **c** Photomicrographs of juxtacellularly labeled and immunhistologically identified pyramidal, SST and PV neurons ($n = 10$, 6, and 3 independent experiments with similar results for pyramidal, SST, and PV cells, respectively) (bar scale = 30 μm). Bottom: Raster plots representing the activity of the neurons described above in response to 2-min DBS at 120 μA. Spikes occurring during the DBS are represented in red. **d** Heatmaps of individual pyramidal, PV and SST neuron activity (normalized by the maximal firing rate of each neuron), before, during and after DBS (10s bins) (only non-silent neurons are represented). **e** Averaged time course of DBS-induced modulation of pyramidal, PV and SST neuron activity (bars: mean ± SEM of all non-silent neurons, and the mean ± 2 × SD of the baseline is superimposed). Boxplots indicate the firing rates before, during and after DBS of pyramidal (Off vs DBS: $p = 0.0297$, and DBS vs Post: $p = 0.0085$, $n = 28$, two-tailed Wilcoxon signed-rank test), PV ($p = 0.0074$ and $p = 0.0495$, $n = 27$) and SST ($p = 0.0040$ and $p = 0.0019$, $n = 26$) neurons (including silent neurons). Box plots: the center is the median; box: 25% and 75% quartiles; whiskers extend to the last data point within 1.5* the interquartile range outside of the box range (data outside the whisker range are not shown, but are included in the statistical analysis). **f** Proportion of pyramidal, PV, and SST neurons activated (white), inhibited (black), non-modulated (gray) during DBS or silent (hatched) throughout the recording. The number of neurons is indicated for each category. *$p < 0.05$; **$p < 0.01$.

and synaptic strengths (Fig. 6a) were guided by experimental[23–29] and modeling[30] data. In particular, we modeled PV and SST interneurons as fast-spiking and low-threshold spiking neurons, respectively. As for the network architecture, we set lower conductances for excitatory synapses compared to inhibitory ones[31,32]. Both PV and SST interneurons inhibited pyramidal cells, with similar connection probability but a stronger synaptic weight from PV to pyramidal cells[25,26]. Importantly, only PV cells received excitatory feedback[24]. The choice of asymmetric inhibitory strength between PV and SST neurons, as well as the respectively weak and strong self-inhibition for SST and PV cells are based on previous reports[26,31]. The hyperexcitability of pyramidal cells in the parkinsonian condition[10] was considered by decreasing their firing threshold. Finally, DBS-mediated currents were modeled as instantaneous excitatory inputs to all cells, with no preconception of the pathway involved to reach each population (directly via antidromic or orthodromic connections from STN to cortex, or indirectly through basal ganglia loops), thus representing the sum of DBS network effects.

Consistent with our experimental findings, we observed a decrease in the firing rate of both pyramidal and PV cells while SST interneurons were activated at all tested frequencies and the impact was stronger with increasing stimulation frequency ($p < 0.001$; Fig. 6a). This phenomenon remained robust to changes in the parameters of 130 Hz DBS stimulation (pulse duration and amplitude; Supplementary Fig. 5a) and linearly scaled with the overall amount of current injected in the network (Supplementary Fig. 5b). The impact of DBS on network activity was almost immediate (15–20 ms lag) since DBS acts as an instantaneous current in our simplified model.

We next evaluated the relative contributions of PV and SST interneurons for driving changes in pyramidal cell activity under DBS. We found a larger decrease of pyramidal cell activity in response to DBS-induced stimulations of SST compared to PV interneurons (Fig. 6b and Supplementary Fig. 5c). In particular, no decrease in pyramidal cell activity was observed in the absence of DBS-induced current on SST interneurons (Fig. 6b and Supplementary Fig. 5c). We hypothesized that the asymmetrical efficacy of cortical interneurons in reducing pyramidal cell firing under DBS could be due to their asymmetrical connectivity with pyramidal cells. Indeed, in the motor cortex, L5 SST interneurons engage in feedforward inhibition, receiving no or little excitation from L5 pyramidal cells[24], in contrast to PV interneurons. This hypothesis was validated when evaluating the impact of adding an excitatory feedback from pyramidal cells to SST neurons on pyramidal cell firing: once the strength of this connection became comparable to the excitatory drive from pyramidal cells to PV neurons, the modulation of pyramidal cell activity under DBS was no longer detectable (Fig. 6c).

Finally, we modeled the opto-activation of cortical interneurons by adding an external current to half of PV or SST neurons, constituted by a series of pulses repeated at the same frequency as those used experimentally. These simulated opto-activations of PV and SST interneurons at 67 and 130 Hz both led to a remarkable inhibition of pyramidal cell activity (Fig. 6d), yielding a very sparse residual activity of pyramidal cells, as experimentally observed (Supplementary Fig. 4c, d).

**Cortical information processing under DBS or opto-stimulation.** Based on our simplified network model, we investigated how pyramidal cell activity reduction, by DBS or SST opto-activation, could account for the improved motor function. The impact of the modifications of the network activity on information transmission in these conditions is not heuristically obvious: in Parkinson's disease, the increased excitability of pyramidal cells could make the system more responsive to stimuli, but the higher spontaneous activity may interfere with stimulus-evoked activity. In contrast, the lower spontaneous activity induced by DBS and DBS-guided opto-activation may avoid signal degradation, but could bring the system to a less responsive state. We first tested whether pyramidal cells conserve the capacity to generate specific patterns of activity in response to stimuli despite sparse activity under DBS or opto-activation. We applied 28 inputs with different motifs (deterministic constant and ramping stimuli, and stochastic Ornstein–Uhlenbeck processes; Supplementary Table 2) and intensities to subsets of pyramidal cells (Fig. 7a, b), and systematically evaluated the information conveyed by the network. We found that for medium to high amplitude stimuli, pyramidal cell responses were less correlated with the stimulus in parkinsonian condition compared to control. Both DBS and SST opto-activation increased the correlation coefficients when compared to parkinsonian condition, with smoother response profiles to each stimulus (Fig. 7b, c and Supplementary Fig. 6). PV opto-activation led to weak correlation coefficients (Fig. 7b, c and Supplementary Fig. 6). Importantly, the differential impact of PV and SST opto-activation remained consistent over a large range of current intensities. The reduced efficacy of PV opto-activation was not only due to the higher silencing level of pyramidal cells but also to the induction of oscillatory-like responses (Supplementary Fig. 7a, b). Finally, low amplitude stimuli yielded weak correlations that were not significantly different across conditions and thus did not confer any advantage to the parkinsonian condition (Fig. 7c).

We then tested how accurately M1 activity patterns could be decoded by downstream areas, such as the striatum. We used four machine-learning decoding algorithms (nearest centroid

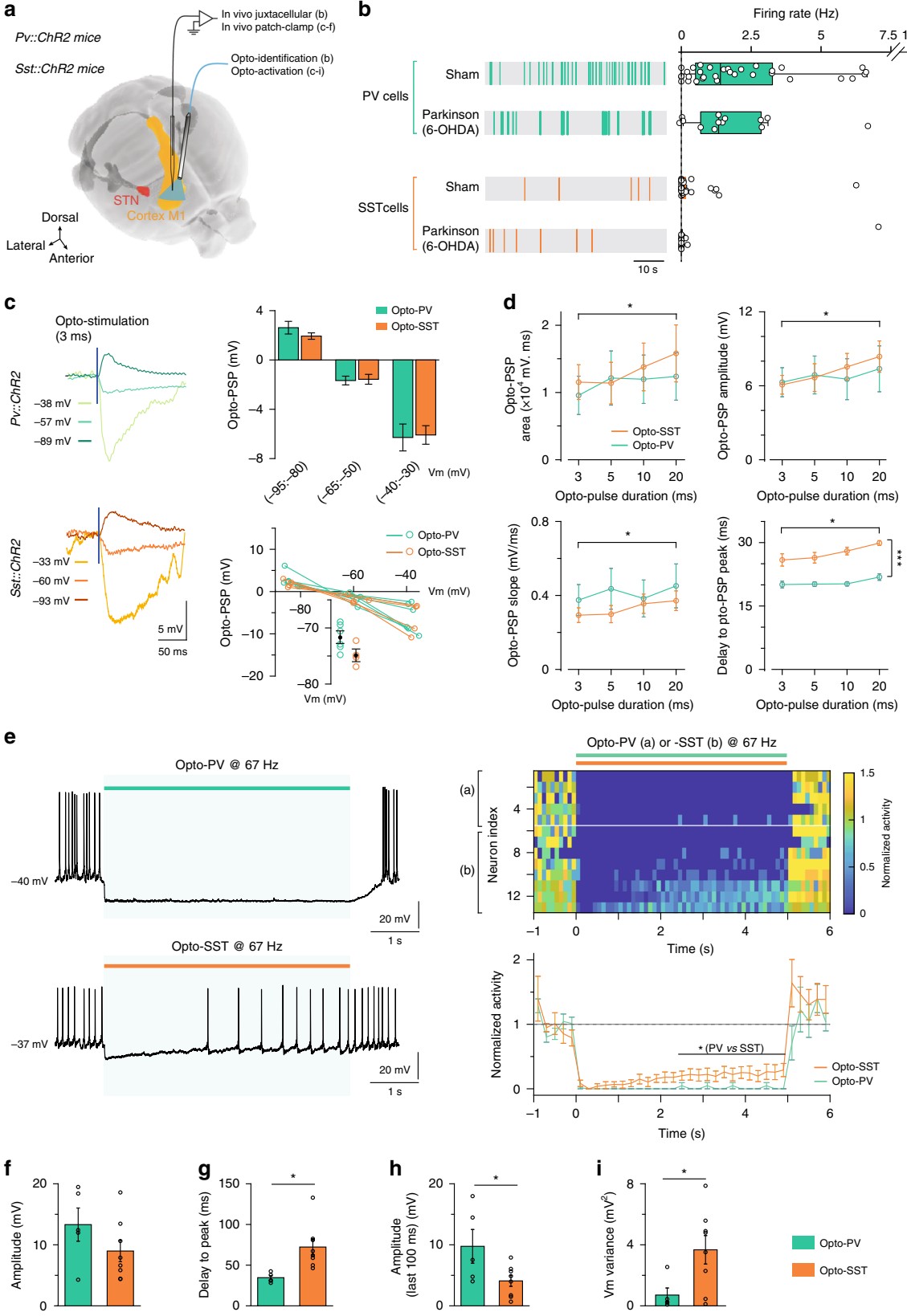

classifier, multinomial logistic regression, linear discriminant analysis and support vector machines; Supplementary Table 3) to discriminate network responses to 20 different stimuli (Fig. 7a, d). Binned spiking patterns from all pyramidal cells were densified using a random matrix to mimic highly convergent cortico-

striatal information transfer and the resulting matrices were used as inputs to the classifiers. The classifier accuracy at determining the stimulus identity was higher in control, DBS and SST opto-activation conditions relative to the parkinsonian condition; this result was consistent across the four classifiers ($p < 0.001$; Fig. 7d)

**Fig. 4 In vivo synaptic transmission of PV and SST cells onto M1 pyramidal cells. a** In vivo experimental setup. **b** Juxtacellular recordings of opto-identified PV and SST interneurons in sham and 6-OHDA-lesioned *Sst::ChR2* and *PV::ChR2* anesthetized mice. Representative raster plots and spontaneous activity of SST and PV interneurons (boxplots and individual neurons) show similar activity in sham and 6-OHDA-lesioned mice (PV: $p = 0.8391$, $n = 32$ cells in sham vs $n = 14$ in 6-OHDA-lesioned mice; SST: $p = 0.2112$, $n = 32$ cells in sham vs $n = 15$ in 6-OHDA-lesioned mice, Wilcoxon unpaired test). Box plots, center: median; box: 25% and 75% quartiles, whiskers: last data point within 1.5* the interquartile range outside of the box. **c–i** In vivo patch-clamp recordings of M1 pyramidal cells in *Sst::ChR2* and *PV::ChR2* anesthetized mice. **c** Representative traces (left) and averaged peak amplitude (top right) of evoked-PSP following single pulse (3 ms) opto-PV or opto-SST. Opto-PV/SST evoked-PSP of similar amplitudes ($p = 0.3397$, $n = 8$ vs $n = 9$, $p = 0.8632$, $n = 7$ vs $n = 7$ and $p = 0.8838$, $n = 8$ vs $n = 7$, for −100, 0 and +100 pA current injection, respectively, unpaired *t*-test). (bottom right) Opto-PSPs voltage-dependency, which reversed at −71.5 for opto-PV ($n = 6$) and −75.2 mV for opto-SST ($n = 4$) ($p = 0.0542$, unpaired *t*-test). Calculated chloride reversal: −71.9 mV. **d** Area, peak amplitude, slope and delay of evoked-PSP by opto-PV ($n = 7$) or opto-SST ($n = 9$). In all cases the pulse duration factor had a significant influence (area: $F_{3,33} = 4.946$, $p = 0.0060$; peak: $F_{3,33} = 6.206$, $p = 0.0029$; slope: $F_{3,33} = 2.956$, $p = 0.0466$; delay: $F_{3,33} = 6.349$, $p = 0.0016$, 2-way repeated measures ANOVA), with no difference between PSP evoked by opto-PV/SST, except for the delay to peak ($F_{1,11} = 66.39$, $p < 0.0001$). **e** Representative voltage traces and heatmap of individual cortical neurons normalized activity, and averaged time course for opto-PV ($n = 5$) or opto-SST ($n = 8$) at 67 Hz (3 ms pulses); Injected current: +100 pA. Opto-PV/SST both inhibit pyramidal cell activity ($p = 0.0392$, $p = 0.0010$ paired *t*-test), with a lower effect of opto-SST (last 2.5 s, $p = 0.0326$, Mann–Whitney *U* test). Peak amplitude ($p = 0.2270$) (**f**), delay to trough ($p = 0.0124$) (**g**), amplitude (last 100 ms) ($p = 0.0384$) (**h**) and Vm variance ($p = 0.0371$) (**i**) of the evoked-hyperpolarization by opto-PV/SST at 67 Hz ($n = 5$ and 8 cells from 4 PV::Chr2 and 6 Sst::ChR2 mice, respectively). All data are presented as mean ± SD. *$p < 0.05$; ***$p < 0.001$. All statistical tests are two-tailed.

and the efficacy of cortical interneurons opto-activation broke down for very high stimulation intensities, once pyramidal cells were too sparsely responsive to each stimulus (Supplementary Fig. 7c).

Thus, our modeling results indicate that reducing pyramidal cell hyperactivity by DBS or preferentially by SST opto-activation enables the cortical network to efficiently encode stimulus content and may facilitate the extraction of this information by down-stream areas. These results are robust to variations in the stimulation parameters (Supplementary Fig. 7) and choices of network parameters. Importantly, the asymmetrical connectivity of SST interneurons with L5 pyramidal cells[24] accounts for their higher efficiency at inhibiting pyramidal cells under DBS, when all three populations receive an external periodic pulse-shaped excitatory current; furthermore, this asymmetrical connectivity has important consequences in structuring the temporal dynamics of pyramidal cells under SST opto-activation as it does not impose an oscillatory-like pattern as PV opto-activation does in our model.

## Discussion

In Parkinson's disease, although DBS has been widely described as acting on distinct basal ganglia nuclei[4–6], recent studies have also pointed towards an impact of DBS at the cortical level[7,8,11–17]. Here, we show that M1 pyramidal cells display a hyperactivity in parkinsonian rodents, which is counteracted by DBS. Our in vivo intracellular and juxtacellular recordings indicate that DBS recruits cortical GABAergic networks. Furthermore, opto-activation of cortical interneurons, and in particular SST inter-neurons, alleviates key motor symptoms in parkinsonian mice. Our theoretical modeling reveals that both DBS and SST inter-neuron opto-activation increase information processing capacity by counteracting the cortical hyperactivity, without interfering with the network temporal dynamics.

As reported here and in agreement with data collected from patients[12–17], hyperactivity of pyramidal neurons may be a neuropathological hallmark in Parkinson's disease. It likely results from abnormal basal ganglia dynamics as well as substantial dopaminergic denervation of the motor cortex possibly altering the excitation-inhibition balance[17]. It has been suggested that the therapeutic actions of levodopa may arise from dampening M1 hyperactivity, since it reduces M1 glutamate levels and increases the inhibitory tone in 6-OHDA-lesioned rats, and reduces M1 blood-oxygenation and glucose metabolism in parkinsonian patients[13,17].

We report that DBS reduces pyramidal hyperactivity and exerts a dualistic action on cortical GABAergic interneurons, namely activation of SST and inhibition of PV interneurons. This agrees with a study showing that in a genetic parkinsonian rodent model, DBS reduced the abnormal M1 activity, hypothesized to result from the activation of cortical GABAergic interneurons[33]. The correlated time courses of the decreased hyperactivity of pyramidal cells and SST activation upon DBS combined with our modeling result suggest that the main beneficial effects of DBS on normalization of cortical activity and processing capabilities seem to be conveyed by the activation of SST cells. Several pathways could account for SST recruitment by DBS: antidromic activation of the cortico-subthalamic fibers, orthodromic activation of subthalamo-cortical fibers, and/or basal ganglia-thalamo-cortical loops[4,11]. The activation of SST interneurons by antidromic axonal reflex is unlikely to occur since cortico-STN pyramidal cells lack collaterals to SST interneurons[24]. The orthodromic STN-cortex pathway could recruit SST neurons, since it mainly targets superficial cortical layers[34], where pyramidal cells directly connect to L5 SST interneurons[35]. Concurrently, cortical effects of DBS could occur via recruitment of other basal ganglia-thalamo-cortical loops. Other interneurons might be engaged during DBS, such as VIP cells[36], since their inhibition would disinhibit SST cells. The concurrent recruitment of these anatomical pathways likely results in a new cortico-basal-ganglia wide state of equilibrium. This DBS-induced state stabilizes over time, which would explain the observed onset and offset delays in pyramidal and SST responses to DBS.

We mimicked DBS effects by opto-activating M1 SST inter-neurons and revealed this was sufficient to alleviate parkinsonian symptoms. Interestingly, both in experimental and modeling data, while opto-activation of PV interneurons was more efficient than SST to inhibit pyramidal cell activity, the beneficial outcome on motor symptoms and cortical processing was weaker compared to SST opto-activation. Recent studies have highlighted the key roles operated by SST interneurons in cortical information processing[35,37–39]. SST opto-activation in the somatosensory cortex ameliorates symptoms in a mouse model of neuropathic pain[40], and improves affective state discrimination in the prefrontal cortex[41]. Under DBS, the need for SST interneuron activation to efficiently reduce cortical hyperactivity stems from their asymmetrical connectivity with pyramidal cells (Fig. 6b, c). Indeed, within L5 motor cortex, PV interneurons are involved in strong feedback loops with pyramidal neurons, while SST interneurons, in part recruited by L2/3 pyramidal neurons, provide disynaptic feedforward inhibition[24]. Furthermore, this asymmetrical connectivity has important

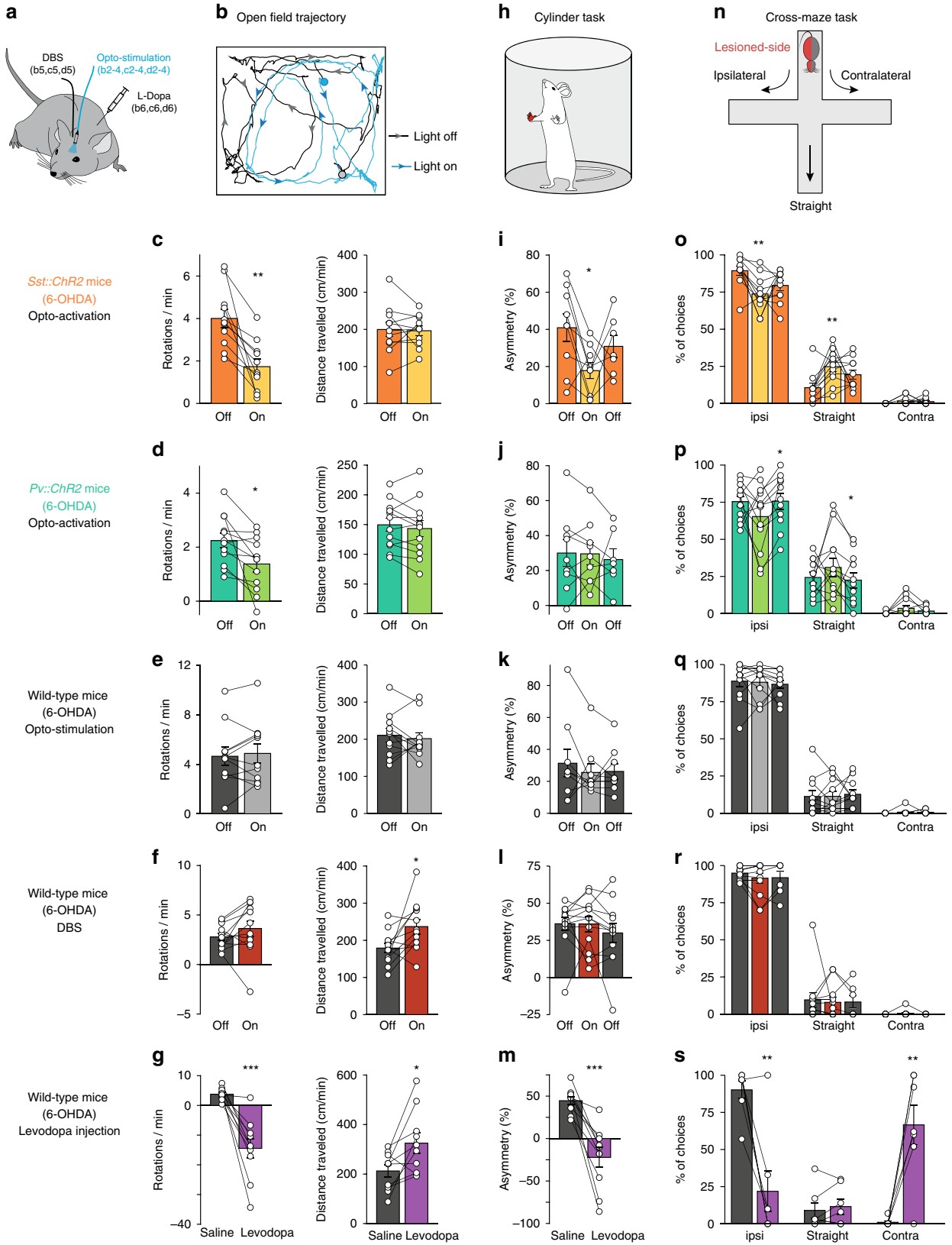

consequences on the temporal structuring of the network activity patterns. Additional mechanisms could further reinforce this effect. The position of SST terminals onto pyramidal dendritic branches, by leading to a direct modulation of excitatory inputs, may explain why SST activation improves information processing compared to PV, which exerts a massive shunting (Fig. 4e, i) through somatic-targeted inhibition[37,42,43]. Overall, SST interneurons could thereby be more efficient than PV cells at exerting a sustainable inhibitory influence by filtering excitatory inputs onto pyramidal neurons (rather than simply silencing pyramidal cell output by shunting the inputs), thus resulting in an ameliorated performance of motor functions. A more detailed model, including a multi-compartmental

**Fig. 5 Opto-activation of M1 SST interneurons alleviates parkinsonian symptoms. a** Unilaterally 6-OHDA-lesioned *Sst::ChR2, Pv::ChR2* and wild-type mice were either implanted with an optic fiber in M1 ipsilateral to the lesion, or implanted with a stimulating DBS electrode in the STN, or injected with levodopa (6 mg/kg). **b–g** Rotational behavior and locomotor activity were quantified in an open field, in the presence or absence of light, DBS or levodopa injection (**b**: 30 s of trajectory shown in both conditions in a *Sst::ChR2* mouse; circles and arrows indicate the starting point and direction of the animal). The number of rotations ipsilateral to the 6-OHDA lesion was decreased during opto-stimulation in *Sst::ChR2* (**c**: $p = 0.0010$, $n = 11$ mice) and in *Pv::ChR2* (**d**: $p = 0.0123$, $n = 12$) mice, but not in wild-type (non-opsin expressing) animals (**e**: $p = 0.5591$, $n = 11$). DBS did not affect the rotational behavior (**f**: $p = 0.2082$, $n = 12$), while levodopa treatment induced contralateral rotations (**g**: $p = 0.0004$, $n = 10$). Opto-stimulation did not affect the distance traveled in either mouse strain (*Sst::ChR2* $p = 0.5721$; *Pv::ChR2* $p = 0.0737$; wild-type $p = 0.6289$), while both DBS and levodopa treatment increased the locomotor activity ($p = 0.0438$, and $p = 0.0135$, respectively). **h–m** Cylinder test: (**h**), the asymmetry in front paw usage preference decreased when SST interneurons were opto-activated (**i**: $p = 0.0156$, $n = 9$), but not when opto-stimulation was used in *Pv:ChR2* (**j**: $p = 0.9139$, $n = 9$) or wild-type mice (**k**: $p = 0.38$, $n = 9$). DBS had no effect on front paw usage (**l**: $p = 0.9768$, $n = 12$), while levodopa injection reversed the front paw bias (**m**: $p = 0.0002$, $n = 10$). **n–s**, Cross-maze test: (**n**), ipsilateral turn preference was decreased in favor of straight turns during opto-stimulation in *Sst::ChR2* mice (**o**: ipsi turns $p = 0.0019$, straight turns $p = 0.0022$, and contra turns $p = 0.1014$, $n = 11$), but not in *Pv::ChR2* (**p**: $p = 0.1876$, $p = 0.3419$, and $p = 0.0960$, $n = 12$) nor in wild-type (**q**: $p = 0.8292$, $p = 0.9755$, and $p = 0.3409$, $n = 11$) mice. DBS did not affect ipsilateral turn bias (**r**: $p = 0.2040$, $p = 0.7675$, and $p = 0.3388$, $n = 12$), while levodopa reversed the preference toward a majority of contralateral choices (**s**: $p = 0.0029$, $p = 0.4072$, and $p = 0.0029$, $n = 7$). Mean ± SEM of all tested mice, as well as individual mice, are represented for all tests. All statistical tests are two-tailed paired *t*-tests. *$p < 0.05$; **$p < 0.005$; ***$p < 0.001$.

description of pyramidal cells as well as short-term plasticity properties, would allow exploring how these differences in PV and SST physiology could account for the different behavioral outcomes of their opto-activation.

We compared the behavioral effects of SST opto-activation with typical treatments of parkinsonian symptoms: DBS and levodopa. In our unilateral model of parkinsonism, levodopa has strong effects, in particular on the asymmetrical symptoms, where the ipsilateral bias is not only reduced but reversed, thus creating a strong contralateral bias. This is in line with previous studies[44] and thought to result from the supersensitivity of ERK signaling in the 6-OHDA-lesioned hemisphere[45]. Our DBS results mirror a recent study in 6-OHDA-lesioned mice[22], reporting improved locomotion with no effect on rotational bias. Surprisingly, DBS in mice does not recapitulate the typical motor improvements observed in rats[8,46,47], although results in rats also vary depending on the intensity and the duration of stimulation[48–50]. Several studies, including ours, choose an intensity below the dyskinetic/dystonia threshold, as used in clinical DBS. In mice, dyskinesia is reported in the absence of effect on rotational bias over a large range of parameters studied[22]. The small size of the mouse STN could make it difficult to stimulate efficiently enough to reduce asymmetry without deleterious impact on the pyramidal tract[51]. Importantly, opto-stimulating cortical interneurons avoids this issue, thus explaining why SST opto-activation was efficient on asymmetrical behavior. Yet unaffected hypolocomotion under opto-activation suggests that non-cortical effects of DBS (i.e. direct modulation of basal ganglia structures[4–6], unlikely to be mimicked by cortical opto-activation) also contribute to the beneficial locomotor effects. Alternatively, the magnitude or spatial extent of DBS cortical effects (milder stimulation of SST neurons reaching potentially larger regions of frontal cortex than single fiber opto-stimulation) could contribute to thedifferent behavioral outcomes observed. Importantly, M1 SST opto-activation decreases asymmetrical behavior in 6-OHDA-lesioned mice but has no effect in sham animals. This argues for a therapeutic effect, rather than a net sum of two unrelated biases (pathological ipsilateral bias, and physiological contralateral bias from the inhibition of ipsilateral M1). Yet, the similar firing of interneurons in (anesthetized) sham and 6-OHDA animals suggests that this therapeutic effect does not stem from a pathological hypoactivity of interneurons in parkinsonian conditions, though, anesthesia effects on SST firing patterns could occlude differences between sham and lesioned animals. Similarly, our data pointing to SST recruitment by DBS was obtained in anaesthetized rodents. While we can reasonably hypothesize that in the absence of the dampening effect of anesthesia,

DBS-activation of SST interneurons would be even higher, hence exerting an even stronger inhibition onto the hyperactive pyramidal cells, this remains to be confirmed.

The methodology used for testing the information processing capabilities of the network (Fig. 7) could be applied to experimental recordings: differences in the correlation between specific motor patterns and large-scale in vivo M1 single-unit activities as well in the responses reliability over multiple trials could be found by comparing sham, 6-OHDA lesioned mice, with or without DBS or SST opto-activation. Moreover, since the segregation of sensorimotor maps is degraded in the basal ganglia in the parkinsonian condition[52,53], neuronal decoding algorithms trained on a subset of trials in which different types of movements are initiated could test our model prediction, i.e. that DBS or SST opto-activation would restore neuronal selectivity by decreasing cortical hyperactivity.

Overall, our study reveals the cortical SST interneurons as a promising therapeutic target for Parkinson's disease. Therefore, increasing the inhibitory drive into the motor cortex could represent a useful strategy to improve motor symptoms. While there is an emerging therapeutic potential of optogenetics[54–60], its medical application is still in its infancy, with the first human clinical trial underway to treat retinitis pigmentosa (clinicaltrials. gov, NCT03326336 and NCT02556736). More work is necessary to fully explore and optimize safe transfection and light delivery in humans[58,60,61]. Nevertheless, targeting cortical GABAergic networks with pharmacology or non-invasive brain stimulation[61] such as transcranial stimulation[62] could provide less invasive strategies than DBS, thus benefiting a larger population of parkinsonian patients.

## Methods

**Animals**. All experiments were performed in accordance with the guidelines of the local Ethics Committee (CEEA-59) and EU directive (2010/63/EU). Adult male OFA rats ($n = 31$, 175–200 g) (Charles River, L'Arbresle, France) were used for electrophysiology. Adult mice of both sexes ($n = 158$, 2–12 months) were used for unitary extracellular recordings combined with opto-tagging, behavior, and immunostaining: wild type C57Bl6 (Charles River), and hybrid transgenic *Pv:: ChR2* and *Sst::ChR2*. The hybrid transgenic mice were heterozygous for both genes, obtained by mating a cre-driver transgenic line ensuring specific expression of the cre recombinase in PV or SST neurons (PV-cre: Jackson Laboratory stock #008069 or SST-cre: Jackson Laboratory stock #013044, Charles River) with a transgenic reporter line containing channelrhodopsin-2(H134R) (ChR2: Jackson Laboratory stock #012569, Charles River). Animals were housed in an approved animal facility under standard 12-hour light/dark cycles, with food and water available ad libitum, and nesting materials provided.

**Rodent model of Parkinson's disease: 6-OHDA lesions**. Rats were anesthetized with pentobarbital (30 mg/kg, i.p.), and adjusted to a surgical plane with ketamine

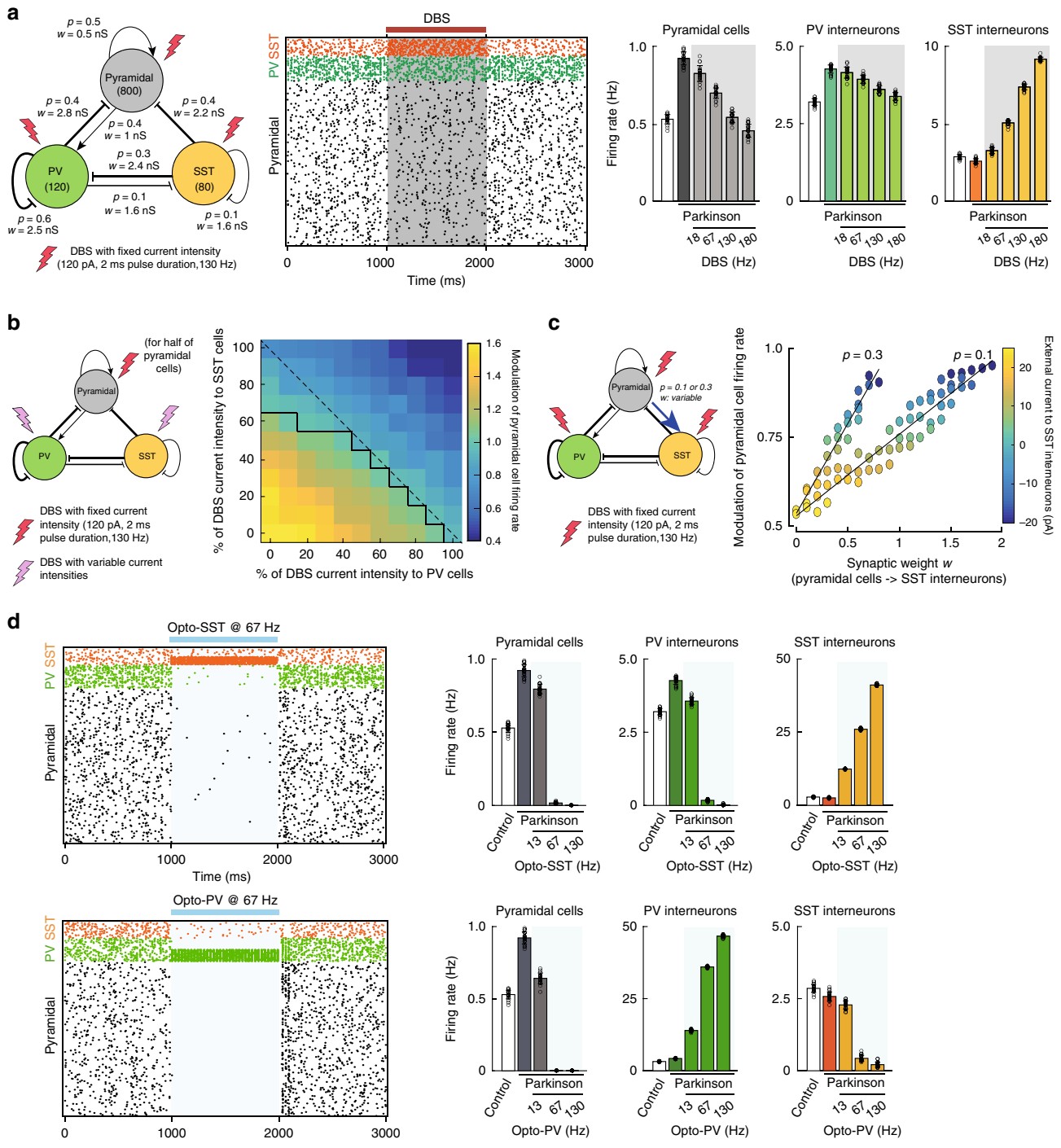

**Fig. 6 A simplified M1 model recapitulates DBS effects on cortical activity. a** Model architecture (left) of layer V motor cortex, with population-specific connection probability $p$ and synaptic weight $w$ (arrow: excitatory, bar: inhibitory). Raster plot (center) before, during, and after DBS. Population firing rate (mean ± SD) across different conditions and for varying DBS frequencies (right). All firing rates are significantly ($p < 0.001$) different from those observed in the parkinsonian condition ($p < 0.001$, $n = 20$ independent simulations of 1000 ms, with different intrinsic noises and connectivity patterns). **b** Average firing rate of pyramidal cells relative to the parkinsonian condition ($n = 20$ independent simulations) as a function of the percentage of DBS current intensity received by SST and PV interneurons (default DBS parameters with maximal intensity: 120 pA). 50% of pyramidal cells receive DBS current. Black line: isocline corresponding to a ratio of 1 (no change from parkinsonian condition). **c** Impact of adding an excitatory feedback connection (with a probability $p = 0.1$ or 0.3 and variable weight $w$) between pyramidal and SST neurons on the modulation of pyramidal cell firing rate under DBS (default DBS parameters). The increase in the excitability of SST interneurons was compensated by decreasing their external current $I_{ext}$ such that the ratio of SST firing rate under DBS vs. parkinsonian condition remained bounded between −0.5 and +0.5 of the initial ratio, obtained without the feedback connection. **d** Raster plot and population firing rate (mean ± SD, $n = 20$ independent simulations) under optogenetic activation of SST (top) and PV (right) interneurons at various frequencies (50% of PV or SST interneurons are opto-activated with pulses of 3 ms, of 600 pA). For panels **a**, **d**: see Supplementary Data 1.

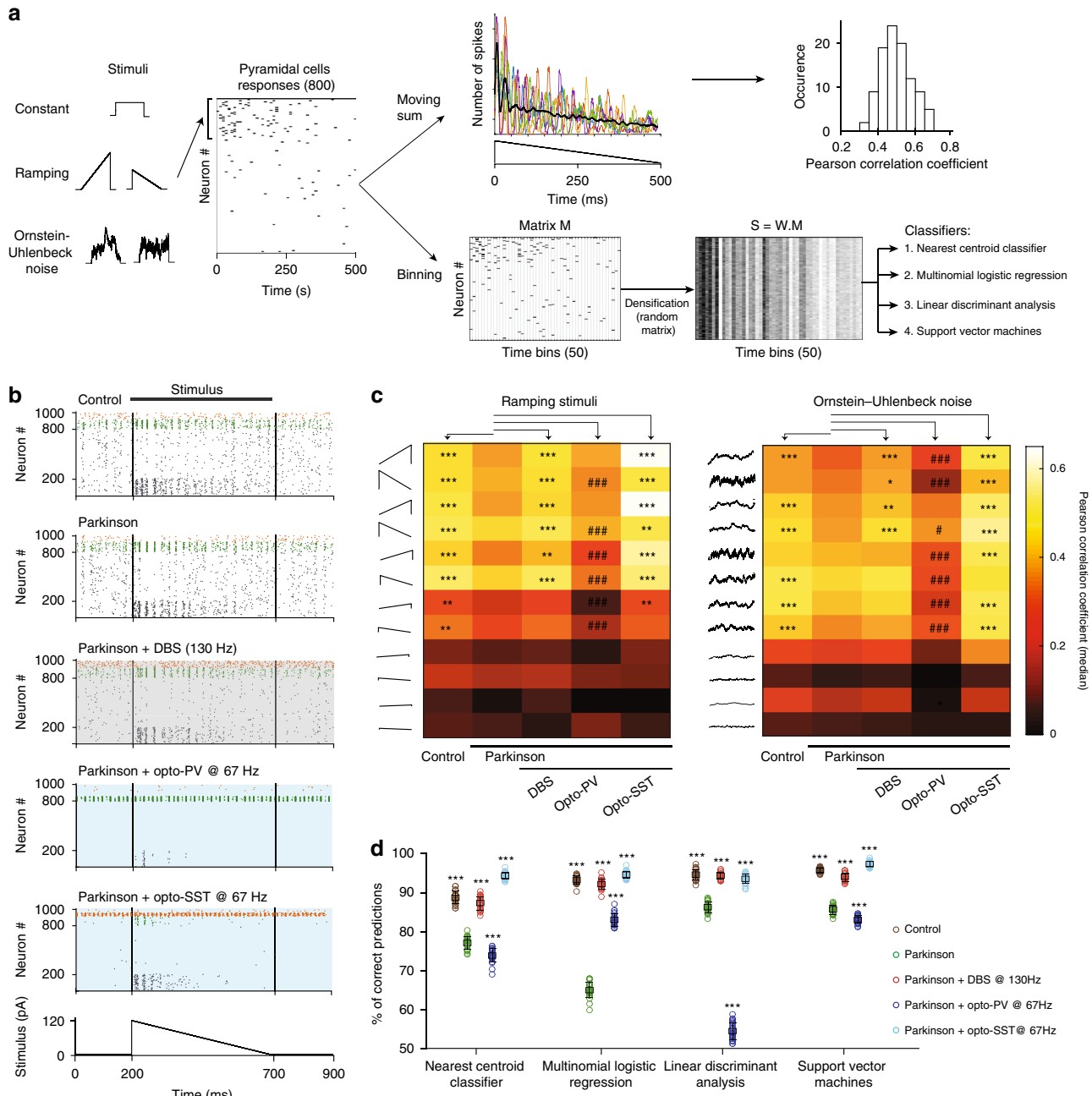

**Fig. 7 Theoretical impact of DBS and opto-PV/SST on cortical information processing. a** Procedure for calculating Pearson correlations between network responses and time-varying stimuli (4 constant stimuli, 8 ramping stimuli, and 8 stochastic stimuli) and for training the classifiers. Example traces of moving spike counts and the resulting distribution of correlation coefficients are displayed on the top right. Binned pyramidal cell spiking responses were first densified using a random matrix to serve as inputs for the classifiers (cf. "Methods"). **b** Raster plots of network responses to a ramp. **c** Heatmaps of median Pearson correlation coefficients ($n = 100$ independent simulations) for ramping stimuli (left) and stochastic inputs (right). **d** Accuracy (mean ± SD, $n = 25$ independent simulations) of four different classifiers, defined as the percentage of correct predictions of stimulus identity from pyramidal cell activity. Statistical comparisons to the parkinsonian condition are performed using One-Way ANOVA (or Kruskal–Wallis test for **c**) with Tukey–Kramer post-hoc tests, *$p < 0.05$; **$p < 0.01$; ***$p < 0.001$. Statistical tests are all two-tailed. For panels **c**, **d**: see Supplementary data 1.

(27.5 mg/kg, i.m.; Imalgène, Mérial, Lyon, France). Mice were anesthetized with an i.p. mix of ketamine (67 mg/kg) and xylazine (13 mg/kg, Rompun, Bayer, Puteaux, France). All animals received a bolus of desipramine (Tocris, Bio-Techne, Lille, France) dissolved in saline (rats: 25 mg/kg, i.p.; mice: 20 mg/kg, i.p.) 30 min before the injection of 6-OHDA (Sigma, Steinheim, Germany) to prevent neurotoxin-induced damage of noradrenergic neurons. For rats, 7 μL of vehicle (saline containing 0.01% w/v ascorbic acid, Sigma) or 6-OHDA (2.5 mg/ml in vehicle) was injected at 16 μL/h in the substantia nigra *pars compacta* (from the interaural line: AP 3.7 mm, ML 2.1 mm, 7.55 mm depth from the cortical surface). For mice, 0.5 μL of vehicle (saline containing 0.02% w/v ascorbic acid) or 0.5 μL of 6-OHDA (6 mg/ml in vehicle) was injected at 3 μL/h in the right medial forebrain bundle

(from bregma: AP 1.2 mm, ML 1.2 mm, DV 4.9 mm). Mice who failed to display spontaneous ipsilateral rotations (i.e., were seen to explore indifferently towards ipsi- and contralateral sides when observed in their homecage and on the scale during daily weighting, without amphetamine challenge) and weight loss in post-lesion recovery were excluded from the study.

**In vivo intracellular electrophysiological recordings**. After 3 weeks of recovery, we performed intracellular recordings in 6-OHDA-lesioned and sham rats maintained in narcotized and sedated states with fentanyl (4 μg/kg, i.p.; Janssen-Cilag, Issy-Les-Moulineaux, France), immobilized with gallamine triethiodide (40 mg,

i.m., every 2 h; Specia, Paris, France) and artificially ventilated (UMV-03, UNO, Zevenaar, Netherlands). Craniotomies were drilled above M1 (from the interaural line: AP 12.5 mm; ML 3.8 mm) and STN (AP 5.2 mm; ML 2.5 mm). Body temperature was maintained at 36.5 °C with a homeothermic blanket. Intracellular recordings were performed using glass micropipettes filled with 2 M potassium acetate (40–70 MΩ) and the active bridge mode of an Axoclamp-2B amplifier (Molecular Devices, Union City, CA). Data were sampled at 25 kHz via a CED 1401 interface using the Spike2 data acquisition program (Cambridge Electronic Design, Cambridge, UK). The STN ipsilateral to the recorded pyramidal cells was stimulated with a bipolar electrode (SNE-100; Rhodes Medical Instruments, Woodlands Hill, CA) at 8.1 mm depth. Electrical stimulation consisted in either single-shock STN stimulations or 130 Hz DBS (60 μs, 2–4 V), using a stimulus isolator (DS2A, Digitimer, WelWyn Garden City, UK) driven by a pulse stimulator (Pulsemaster A300, WPI, Hitchin, UK).

**In vivo juxtacellular electrophysiological recordings**

*Rats.* Unilaterally 6-OHDA-lesioned or sham-lesioned rats were anesthetized with chloral hydrate (400 mg/kg, i.p., supplemented by continuous injection delivered at 60 mg kg$^{-1}$ h$^{-1}$). Craniotomies: see intracellular recordings. Single-unit activity was recorded extracellularly in deep layers of M1 (depth range of recorded neurons: 1422–2622 μm) using glass micropipettes (10–15 MΩ) filled with NaCl (1 M) and identified as pyramidal based on their waveform features (with waveform width from depolarization to trough >1 ms; Supplementary Fig. 1a). Single neuron action potentials were recorded using the active bridge mode of a Axoclamp-2A amplifier (Molecular Devices), amplified with a DAM50 (WPI). Spikes were sampled at 10 kHz via a CED 1401 interface using Spike2 (Cambridge Electronic Design). The STN ipsilateral to the recorded pyramidal cells was stimulated as described in intracellular recordings. Neuronal spontaneous activities were recorded for at least 120 s before, during, and after DBS (60 μs, 2–4 V at 130 Hz).

*Mice.* Mice were anesthetized using urethane (1.5 g/kg i.p.). A craniotomy was made over the STN (from bregma: AP −1.8 mm, ML 1.5 mm). Recording electrodes (15-25MΩ) contained NaCl (0.5 M) and neurobiotin (1.5%). Electrode signals were amplified using an Axoclamp-2B amplifier (Axon Instruments), digitized at 25 kHz and stored in Spike2. STN position (depth of ~4.3 mm) was based on characteristic discharges of STN neurons in urethane anesthesia[63]. Bursts in STN neurons were detected using the Poisson surprise method. To confirm STN position, some STN neurons were electroporated and filled with neurobiotin. Once the STN was located, the recording electrode was removed and a bipolar electrode (CBDSF75, FHC, Bowdoin, ME) was targeted into the STN using the same coordinates. For opto-identification of neuronal populations, a craniotomy was made over M1 ipsilateral to the STN (from bregma: +1.5 mm, ML 1.5 mm). An optical fiber connected to the light source (Plexbright 465 nm, Plexon, Dallas, TX) was placed on the cortical surface. Glass electrodes were lowered through the cortex while flashing light pulses (100 ms, 0.5 Hz, 10 mW). After opto-identification, spontaneous activity was defined with 2 min of stable recording before launching DBS (60μs, 120 μA at 130 Hz during 2 min) using pulse isolator and stimulator (IsoFlex + Master-8, A.M.P.I, Jerusalem, Israel). Finally, neurons underwent a series of 1 min opto-activations (3 ms) at 13, 67, and 130 Hz, with at least 1 min of recovery between each series. Neurons were juxtacellularly labeled, and electrolytic lesions of the STN were performed (30 μA DC for 30 s) for posthoc confirmation. Animals with lesions outside the STN were excluded. Because *Pv::ChR2* and *Sst:: ChR2* mouse strains display a small proportion of off-target expression of ChR2 (Supplementary Fig. 2), we used cluster analysis to combine opto-activation and waveform characteristics for a better identification (Supplementary Fig. 3). Principal component analysis (Matlab) was applied to waveform characteristics (time and amplitude of the trough and of the second peak relative to the first peak, on the filtered waveform) and opto-activation success rate. Neurons were clustered using hierarchical and k-means clustering algorithms (Matlab), testing all combinations of the first 2, to all principal components, with 2–7 target clusters. The resulting cluster silhouettes were ranked and the best score was selected (mean silhouette = 0.73, with 2 principal components, 2 clusters, k-means algorithm), resulting in two clearly separated clusters. Morphological and immunohistochemical identity of labeled neurons ($n = 18$) allowed us to identify the two clusters as pyramidal cells and interneurons. Two cells where the morphological and clustered identity did not match were excluded. Opto-response characteristics: the success rate was the percentage of light pulses eliciting at least a spike (occurring during the pulse); the opto-response latency was the time elapsed between the onset of the light pulse and the first spike within the light pulse (mean, SD and CV of the latency were calculated). Neuronal responses to DBS were classified as non-modulated if the variation of their spontaneous activity was <5% compared to baseline activity.

**In vivo patch-clamp recordings.** Mice were anaesthetized with urethane as described for juxtacellular recordings. A 0.5 × 0.5 mm craniotomy was performed to expose M1 (from bregma AP +0.5 mm, ML 1.5 mm). Borosilicate glass pipettes (3.5–5 MΩ) contained (mM): K-gluconate (125), KCl (8), HEPES (10), phosphocreatine (10), ATP-Mg (4), GTP-Na (0.3) and EGTA (0.3) (adjusted to pH 7.35 with KOH). Signals were amplified using an EPC10-2 amplifier (HEKA Elektronik, Lambrecht, Germany) and sampled at 10 kHz, with Patchmaster v2x32 (HEKA Elektronik). ChR2 was activated using a 475 nm-laser diode light source (FLS-475,

DIPSI, Cancale, France) with single light pulses (3–20 ms, 10 mW) at 0.33 Hz, or with 3 ms-pulse trains at 67 Hz during 5 s. The optic fiber was placed on top of the pia with an angle of 15°. Pyramidal cells were recorded at 790–1350 μm from the pia.

**Behavior**

*Optic fiber or DBS electrode implantation.* For opto-stimulated mice, an optic fiber cannula (200 μm core, 0.39 NA, CFML 12U-20, Thorlabs, Maisons-Laffitte, France) was implanted in M1 (from bregma: AP +1.2 mm, ML 1.5 mm, 0.2–0.4 mm depth from the surface). For DBS-stimulated mice: see juxtacellular recordings. After closing the craniotomy with silicone (Kwik-Cast, WPI), the implant was cemented to the skull and protected by a coppermesh circlet or a plastic cylinder. Mice recovered for 10 days before behavioral testing.

*Optogenetic and DBS stimulations.* Blue light (475 nm) was delivered from a laser diode light source (FLS-475, DIPSI) through a rotary joint (FRJ_1x2i_FC-2FC, Doric Lenses, Québec, Canada) connected to the mouse optic fiber cannula, controlled by a Master-9 (A.M.P.I.) delivering 3 ms light pulses (10 mW max at the tip) at 67 Hz. We chose a stimulation frequency (67 Hz) that was close to the opsin tau-off (~15–20 ms[64]). Indeed, higher frequencies such as 130 Hz (the same frequency as electrical DBS) would lose efficiency quickly, while lower frequencies such as 13 Hz, able to entrain all the neurons at each pulse (Fig. S4a–d), might impose pathological-like synchronous activation[65]. DBS stimulation: electrical current (60 μs, 120 μA, 130 Hz). The light/electrical stimulation and the synchronized video were recorded with KJE-1001 (Amplipex, Szeged, Hungary).

*Levodopa treatment.* In the first session, mice were injected with saline (6 μL/g, i.p.), and went through behavioral tests after 30 min in that order: open field, cross-maze, cylinder test, with 5-min rest in their homecage between each test. In the second session, mice were injected (6 μL/g, i.p.) with levodopa (6 mg/kg) with benserazide (3.75 mg/kg) and underwent the same tests in the same order. All the behavioral testing occurred within 30-100 min after the saline or levodopa injection.

*Open field.* Mice were placed in a 40 × 25 × 20 cm arena for 30 s habituation period. For opto- and DBS-stimulation sessions consisted of alternating opto- or DBS-stimulation periods off and on for 1 min, for 10 min. Saline and levodopa sessions lasted for 5 min. Head position was tracked using DeepLabCut algorithm[66], then custom Matlab scripts.

*Cylinder test.* Mice were placed in a glass cylinder (12 cm diameter). For opto- and DBS-stimulated mice, sessions consisted in 3 epochs (before, during and after stimulation) of 10 rearing episodes each. Saline and levodopa sessions consisted in 10 rearing episodes each. The asymmetry of paw contacts on the cylinder surface was measured as % of contacts using paw ipsilateral to the lesion − % of contacts using the contralateral paw.

*Cross-maze.* Mice were left free to explore the cross-maze (arms: 37 × 7 cm) until they performed at least 30 turns (left, right or straight) within 30 min. Right turns: turns towards the ipsilateral side of the dopaminergic lesion. For opto- and DBS-stimulation, sequential independent sessions were performed: off, on and then off during the whole sessions. For saline and levodopa, 2 sessions (≥1-day interval) were recorded. The percentage of turn in each direction was calculated on the first 30 turns.

**Histology**

*Assessment of 6-OHDA lesion.* Coronal slices (50–80 μm thickness) of striatum and SNc were processed for Tyrosine hydroxylase(TH)-immunohistochemistry, after endogenous peroxidases inactivation, using a rabbit anti-TH (1/1000, 36 h at 4 °C, AB152, Merck Millipore), and biotinylated goat antibody against rabbit IgG (1/200, BA 1000, Vector Laboratories, CliniSciences, Nanterre, France), followed by revelation using an ABC kit (PK-6100, Vector Laboratories).

*Efficacy and specificity of transgenic mouse lines.* Coronal slices (30–50 μm) from *Pv::ChR2* or *Sst::ChR2* mice were processed for immunostaining of PV or SST, and YFP (ChR2 expression reporter). For SST/YFP immunostaining, an extra step of antigen retrieval was performed in citrate buffer for 15 min at 75 °C. Primary antibodies: rabbit anti-PV (1/2500, PV25, Swant, Marly, Switzerland), rat anti-SST (1/100, MAB354, Merck Millipore), chicken anti-GFP (1/1000, AB13970, Abcam, Paris, France), incubated 24 h (PV/YFP) or 72 h (SST/YFP) at 4 °C; and secondary antibodies: donkey anti-rabbit coupled to Alexa647 (1/500, 711-605-152, Jackson ImmunoResearch), donkey anti-rat coupled to Alexa647 (1/500, 712-605-153, Jackson ImmunoResearch), and goat anti-chicken coupled to Alexa488 (1/500, 103-545-155, Jackson ImmunoResearch) incubated 1–2 h at RT.

*Juxtacellular labeling.* Floating coronal sections (60 μm) were incubated (2 h at RT) in Alexa 488-conjugated streptavidin (1/250, S11223, Invitrogen, Thermofisher) in

PBS containing 0.2% Triton X-100. Sections were processed for immunostaining of SST and PV as described above.

Images were taken with a stereozoom fluorescence microscope (Axiozoom, Zeiss) or confocal microscope (SP5, Leica), and analyzed using ImageJ software.

**Statistics**. Unless otherwise stated, normal data are displayed as mean ± SEM, and differences between groups were assessed using two-tailed unpaired or paired $t$-test. Non-normal data are presented as boxplots (Matlab built-in boxplot function defaults) where the center line is the median, the box represents the 25% and 75% quartiles (Q1 and Q3), and the whiskers extends to the last data point within 1.5* the interquartile range outside of the box range (bottom whisker: Q1–1.5* (Q3–Q1), and top whisker: Q3 + 1.5*(Q3–Q1)) (data outside the whisker range are not shown, but are included in the statistical analysis), and differences between groups were assessed using two-tailed Mann–Whitney $U$ test (unpaired data) and Wilcoxon's signed rank test (paired data). Normality of each dataset was tested using D'Agostino and Pearson's Omnibus $K^2$ test. 2-way ANOVAs were performed using Prism 5.0 (GraphPad, San Diego, CA, USA).

**Computational model: spiking network model**. We built a simplified spiking model of layer 5 of the motor cortex: the network consisted of 800 pyramidal, 120 PV and 80 SST cells, consistent with the 1:5 ratio between cortical excitatory and inhibitory cells and the larger proportion of PV vs. SST neurons in L5[25]. Neuron of index $i$ was modeled as an adaptive exponential integrate-and-fire neuron[67]: its activity depends upon the dynamics of a fast voltage variable $v^i$ and a slow adaptation variable $w^i$, according to the Eqs. (1) and (2):

$$dv^i = \frac{1}{C}\left(-g_{\text{leak}}(v^i - E_{\text{leak}}) + g_{\text{leak}}\Delta_{\text{thres}}\exp\left(\frac{v^i - V_{\text{thres}}}{\Delta_{\text{thres}}}\right)\right.$$
$$\left. - w^i + g^i_{\text{exc}}(E_{\text{exc}} - v^i) + g^i_{\text{inh}}(E_{\text{inh}} - v^i) + I_{\text{ext}} + I_{\text{DBS}} + I_{\text{stimulus}}\right)dt \quad (1)$$
$$+ \frac{\sigma}{\sqrt{\tau_m}}d\xi^i$$

$$\tau_w \frac{dw^i}{dt} = a(v^i - E_{\text{leak}}) - w^i \quad (2)$$

$$\tau_{\text{exc}}\frac{dg^i_{\text{exc}}}{dt} = -g^i_{\text{exc}} + \tau_{\text{exc}}\sum_j \sum_{\{t_{\text{exc}}\}} w^{ji}_{\text{exc}}\delta(t - t^{j,i}_{\text{exc}} - \text{delay}) \quad (3)$$

$$\tau_{\text{inh}}\frac{dg^i_{\text{inh}}}{dt} = -g^i_{\text{inh}} + \tau_{\text{exc}}\sum_j \sum_{\{t_{\text{inh}}\}} w^{ji}_{\text{inh}}\delta(t - t^{j,i}_{\text{inh}} - \text{delay}) \quad (4)$$

The value of the voltage $v^i$ is determined by intrinsic dynamics and input integration mechanisms. The intrinsic dynamics consists of a linear relaxation towards the resting state and an exponential spiking current active once the spike threshold $V_{\text{thres}}$ is reached. The voltage also integrates excitatory/inhibitory synaptic inputs from the network as well as external stimulations: external constant currents $I_{\text{ext}}$, specific stimuli $I_{\text{stimulus}}$, DBS-induced currents $I_{\text{DBS}}$, as well as noisy currents modeled as independent Gaussian white noise ($\xi^i$). These stochastic inputs encompass the variety of sources of fluctuations of the voltage[68], whose amplitude is determined by the parameter $\sigma$. The slower adaptation variable $w^i$ serves as a negative feedback, whose dynamics is linear in $v^i$ and controlled by the parameter $a$ determining the strength of the adaptation-voltage coupling.

A spike is triggered when the voltage approaches the threshold $V_{\text{thres}}$. Upon firing, the variable $v^i$ is reset to a fixed value, equal to the leak reversal potential, $V_{\text{reset}} = E_{\text{leak}} = -60\text{mV}$[67], whereas $w^i$ is increased by a fixed amount $b$, corresponding to the spike-triggered adaptation.

Choices of intrinsic parameters (Supplementary Table 1) were guided by experimental data. First, pyramidal cells had a larger leak conductance $g_{\text{leak}}$ and capacitance $C$ compared to interneurons. The spike threshold of pyramidal cells was adjusted in parallel with the amount of constant external current $I_{\text{ext}}$ received such that, in absence of stimulus, their firing rate remained around 1 Hz[23]. Secondly, PV cells were modeled as fast-spiking interneurons, with a sharp spiking onset and no adaptation[25] whereas the profile of SST interneurons was characterized by a lower spike threshold and the existence of spike-frequency adaptation[26,67,69]. Adaptation parameters were chosen following Brette and Gerstner[67] and Naud et al.[69]. The parkinsonian model incorporated the hyperexcitability of pyramidal neurons through a decreased spike threshold[10].

Synaptic connections were randomly distributed, with a probability $p$ and fixed synaptic weight $w$ (Fig. 6a). We opted for a simple conductance-based description of synaptic currents: the excitatory and inhibitory conductances $g_{\text{exc}}$ and $g_{\text{inh}}$ display a discrete jump following spikes (arriving at times $t_{\text{exc}}$ or $t_{\text{inh}}$), after a delay of 1 ms, and decay exponentially with time constants $\tau_{\text{exc}}$ and $\tau_{\text{inh}}$ and reversal potentials $E_{\text{exc}}$ and $E_{\text{inh}}$, respectively[50] (Eqs. (3) and (4)).

To mimic the somatic impact of DBS on cortical neurons, we added an external current $I_{\text{DBS}}$ to the equation of the voltage variable in every cell of the network. More precisely, considering the periodic nature of DBS-induced somatic currents, $I_{\text{DBS}}$ corresponds to a series of square pulses (2 ms and 120 pA, at 130 Hz). Considering the different putative activation pathways, $I_{\text{DBS}}$ impact neurons after

specific delays, chosen heterogeneous: 0 ms for half of pyramidal and PV cells (fast antidromic pathways), 2 ms for the other half of pyramidal and SST cells (orthodromic loops). The absence of such delays does not compromise any of the modeling results. For optogenetic stimulations, a series of square pulses (3 ms, 600 pA) was applied to half of PV or SST neurons. This fraction was chosen considering the fact that approximately 70% of targeted neurons express the channelrhodopsin, among which about 70% respond to the light pulse. These optogenetic-like pulses were repeated at a variable frequency, which unless stated otherwise was set to the experimental value of 67 Hz. Additional tests with different current amplitudes, ranging from 200 to 800 pA were also simulated (Supplementary Fig. 7).

Simulations of the network activity were done using a custom code developed in Matlab R2016 (The Mathworks).

**Computational model: analysis**

*Firing rates*. Average firing rates were computed for each population from 1 s simulations of network activity. When mimicking PV or SST opto-activation, the average firing rates of PV or SST interneurons were calculated only based on the neurons that were directly activated.

Heatmaps of pyramidal cell firing rate modulation were obtained by varying the intensity (from 0 to 120 pA) of $I_{\text{DBS}}$ (with default parameters: square pulses of 2 ms, repeated at 130 Hz) received by two populations at the same time, while keeping the default intensity (120 pA) for the third population. Pyramidal cell firing rate modulation corresponds to the average firing rate of pyramidal cells under DBS, divided by the average firing rate of pyramidal cells in the parkinsonian condition.

A linear regression of pyramidal cell firing rates as a function of the amount of $I_{\text{DBS}}$ (corresponding to the total amount of current in pA injected over one second, equal to: stimulation intensity x pulse duration x stimulation frequency) was performed.

*Pearson correlation coefficients*. To test the capacity of the network to discriminate and respond specifically to various inputs, an additional current was injected to a subset of pyramidal cells (200 randomly chosen cells). We explored the responses of the network to three types of stimuli, presented during 500 ms: constant input, linear ramping input (decreasing or increasing amplitude with time) or noisy inputs (Ornstein–Uhlenbeck processes $x_t$ with different mean μ, variance σ and time constant τ, according to $dx_t = \frac{(\mu - x_t)}{\tau}dt + \sigma\,dW_t$ with $x_0 = 0$). These either deterministic or stochastic inputs (Supplementary Table 2) were chosen such as to mimic some activity patterns observed in pyramidal neurons in the motor cortex[70].

For each condition, we used two different methods for quantifying the capacities of extracting information from the network activity patterns. We first measured for each trial the Pearson correlation coefficient ρ between the moving spike count of all pyramidal cells across time for ramping and stochastic stimuli. The moving spike count was calculated based on the sum of the number of spikes across all pyramidal cells, with a moving time interval of 10 ms. The first and last 10 ms of stimulus presentation were discarded to avoid boundary effects. For each stimulus, an average correlation coefficient was obtained by averaging the Pearson correlation index across 100 independent trials (with different intrinsic noises, but identical connectivity matrices, to explore specifically the variability of the responses of a given network to the same input).

*Classification of stimuli*. We also estimated the efficiency with which downstream neurons might discriminate the network responses to various stimuli, beyond the sole knowledge of the mean firing rate. We used four supervised-learning algorithms (Supplementary Table 3) to classify the responses of pyramidal cells to 20 stimuli. Our approach consisted of the following steps (Fig. 7a): our network responses are a time-binned matrix M in which each row corresponding to a given pyramidal cell contains the number of spikes emitted for every 10 ms. The first 200 rows corresponded to the responses of the pyramidal cells directly activated by the stimulus. In order to reproduce the highly convergent cortical motor inputs received by striatal neurons, we densified the responses by contracting these 800 × 50 M matrices into a 100 × 50 S matrix, defined as: S = W.M, where the weight matrix W is a random matrix, identical for all stimuli presentations, with each element generated from the uniform distribution on the interval [0, 1] (Fig. 7a). These S matrices were then used as inputs for the supervised-learning algorithms (nearest centroid classifier, multinomial logistic regression, linear discriminant analysis and support vector machines[71,72]). For each condition, the dataset consisted of 100 repetitions for each of the 20 stimuli (with the same network parameters and convergence matrix W, but independent realizations of the intrinsic noise ξ). The classifiers were trained to discriminate the population response given the stimulus on 80% of the data sample, using stratified k-fold cross validation ($k = 5$). This procedure was repeated using 5 independent seeds. The training and testing of the classifiers were run using scikit-learn and keras packages in Python 3.5 (Python Software Foundation, www.python.org).

*Statistics*. Normally distributed data are displayed as mean ± SD, and differences between conditions were assessed using one-way ANOVA for each feature and Tukey–Kramer post-hoc test. Since some distributions of correlation coefficients were not normal, all data is summarized using the median and presented in details

as boxplots (using the same convention as described above; Supplementary Fig. 6) and differences between conditions were assessed using the Kruskal–Wallis test (with Tukey–Kramer post-hoc test) on every stimulus. The statistics for the Pearson correlation coefficients and classifier accuracies were performed considering the full model (Supplementary Fig. 7).

**Reporting summary**. Further information on research design is available in the Nature Research Reporting Summary linked to this article.

## Data availability
The data sets generated during and/or analyzed during the current study are available from the corresponding author on reasonable request.

## Code availability
Custom Matlab codes used for the computational model are publicly available on GitHub using the following link: https://github.com/cpiette95/Information_processing_DBS_Cortex.

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

## Acknowledgements

We thank the L.V. lab members, Christian Giaume and Pierre Magistretti for helpful suggestions and critical comments. We thank Sylvie Perez, Alessandra Romei, Farah Hadj-Idris, Angèle Roudeau and Charlotte Branco for technical assistance for 6-OHDA lesioned-rodents and immunohistochemistry. We thank France Maloumian for her help in the schematic illustrations in Figs. 3 and 5. This work was supported by grants from Fondation de France (Maladie de Parkinson) CRCPEN, Fondation du Collège de France, France Parkinson, Fondation Patrick Brou de Laurière, INSERM, Collège de France and CNRS. C.P. is supported by Ecole Normale Supérieure, G.G by Fondation Patrick Brou de Laurière, and S.V., W.D. and M.V. by the Collège de France.

## Author contributions

Conceptualization: L.V., B.D., and J.T.; S.V., B.D., A.A.A., and W.D. performed in vivo extracellular electrophysiological experiments; S.V. performed in vivo juxtacellular electrophysiological, opto-identification and immunohistochemistry experiments; B.D. performed in vivo intracellular electrophysiological experiments; W.D. performed in vivo patch-clamp recordings; M.V. and GG performed in vivo optogenetic, behavioral and immunohistochemistry experiments; M.V., B.D., S.V., C.P., W.D., G.G., and L.V. performed analysis; B.D., S.V., G.G., and M.V. performed 6-OHDA lesions; J.T. and C.P. carried out the conception and the design of the mathematical model; C.P. performed the acquisition and analysis of data from the mathematical model; L.V., B.D., J.T., C.P., M.V., and S.V. wrote the paper and all authors have edited and corrected the paper; Funding acquisition: B.D. and L.V.; L.V. supervised the whole study.

## Competing interests

The authors declare no competing interests.
