## [Peer Review File · Nature Communications]

Reviewers' Comments:

Reviewer #1:

Remarks to the Author:

Paper by Vandecasteele and coworkers investigated the cortical effect of subthalamic nucleus-DBS in animal models of Parkinson's disease. Authors found that DBS rescued the pathological hyperactivity of motor cortex pyramidal cells. Specifically, authors find a selective involvement of somatostatin interneurons (SST). Finally by means of an optogenetics approach, authors demonstrated a beneficial effect of cortical modulation in alleviating motor symptoms in model of Parkinson's disease. Paper is clear and well written and technically advanced. Statistical and computing approach is accurate.

The following point require attention.

- Protocol for stimulation of Chr2 in optogenetics experiments has been reported as 3ms 67 Hz. This frequency of stimulation seems to fast for ChR2 kinetic. Did the authors test it by an in-vitro input-output curve? Use of this protocol should be supported by references or discussed it.
- Authors report a reduction in firing rate of pyramidal neurons recorded with intracellular technique. Are the authors measured spontaneous firing activity? All recorded neurons were spontaneously active?
- Figure 1b: is not clear how frequency is represented in this image.
- Figure 1f is not particularly clear. A better representation should be provided.
- Does the STN stimulation affect other cellular properties of pyramidal neurons, as I_h current or AHP current?
- Effect of DBS stimulation onto cortical firing activity was transitory and or reversible?
- DBS reduce firing activity of PV interneurons but increase firing activity of SST interneurons. According with the normalized firing activity the overall GABAergic input onto pyramidal neurons seems to be quite unaltered. Have the authors any data about the synaptic inhibitory input onto pyramidal cells? Are the interneurons activity downregulated in parkinsonian models? In vitro experiments could help to evaluate this point
- Authors should provide details about reversal potential of the CI calculated for the intracellular solution used to perform experiments.
- PV interneurons stimulation by opto-activation only partially reproduce the improvement in motor symptoms induced by SST opto-stimulation. How authors justify this results? A simple higher inhibition seem not enough to rescue motor symptoms and potential differences in strength of synaptic input could be supposed.
- Paper shed light to an interesting topic, however the overall interpretation seems oversimplified: Although cortical implications seem clear, authors do not consider others potential effect of DBS onto others basal ganglia structure (STN globus pallidus). A better consideration of this aspect would improve the paper.

Reviewer #2:

Remarks to the Author:

The manuscript from Vandecasteele et al. describes a series of experiments that characterize how the cortical microcircuit in M1 responds to deep brain stimulation of the subthalamic nucleus (STN-DBS). The main finding is that STN-DBS is associated with elevated activity of SST interneuron in M1, which the authors show is a contributing factor to the therapeutic effect of STN-DBS in rodents. There are several strengths for this paper. The authors have presented high-quality data. The writing is crisp and clear. The experiments make sense, and the proper controls have been performed. As someone who is not familiar with the STN-DBS or Parkinson's disease literature, I cannot speak to the novelty of the findings. However, I believe on the strengths of the data alone – because it is rigorously collected and using contemporary electrophysiological, optogenetic and behavioral techniques in rodents – this paper is a strong contribution.

I have several comments that may improve the clarity of the presentation. It would be helpful to address the comments, but whether the authors address them would not affect my overall opinion of the paper.

Minor comments:

- It would be helpful to know from the main text for experiments associated with Figure 1a, whether the in vivo juxtacellular recording was done in anesthetized or awake rat. Also it is unclear how the authors know the recorded cortical cells are layer 5 pyramidal neurons. Is this based on spike waveform alone or is there some staining with the juxtacellular recording?

- Figure 1b: Should show data points for individuals in addition to summary bar graphs

- Figure 1c and 1d: The main text and figure could be more clear in terms of when the DBS was applied – when did it start and for how long relative to the recording?

- The electrophysiology was done in anesthetized rats, whereas behavior is done with awake mice. As the authors stated, prior studies have noted that anesthesia substantially influence the activity patterns of SST interneurons. There should be some discussion on this caveat when trying to use the electrophysiology to draw inference about the behavioral results, or vice versa.

- The authors showed that STN stimulation increases the activity of SST interneurons. Is this due to axons from STN neurons to SST interneurons? Or is the activation due to DBS recruiting neighboring regions or axons passing by STN? Even if there is no data, some discussion of the possibilities would be helpful.

- Figure 2d: A few features are interesting in this PSTH: that the recruitment of SST activity increases over time during DBS, and that the SST activity remains heightened for another ~30 s after DBS ended. Any idea why this is the case?

- For Figure 3, the PV interneuron activation is an important comparison. The authors have done this experiment but the results are buried in Extended Data Fig. 4. Either outcome for the PV manipulation experiments - whether that help to restore or has no effect on behavior – is interesting. PV neurons were studied along with SST cells at all stages of the studies, from electrophysiology to behavior to modeling, so it is odd to leave it out of the main figures for behavior. The results should be part of the main figures. To a lesser degree, the results from stimulation of SST interneurons on sham mice are also interesting, and should also be moved to be the main figures. I know they don't add nicely to the 'story', but they are intriguing observations that could lead to future studies.

- For the model, the analysis of injecting various waveforms, and then comparing the fidelity of the representation is quite innovative and appropriate. I have, however, a few questions about the construction of the in silico circuit. Why is the STN-DBS modeled as an excitatory input to all cell types – what justifies this? Perhaps more importantly, do the connection strengths and probabilities in the simulation resemble physiological values? With so many connections, it seems that a fit that matches the observed STN-DBS effects is going to be possible, but it is more a matter of whether the fitted circuit is even close to physiology.

- The model is probably too simplistic because it treats pyramidal neurons as a pool of point neurons. It is probable that the main effect of STN-DBS is a shift in somato-dendritic axis of inhibition. For example, in parkinsonian animals, there may be abnormally high levels of excitatory inputs, and the SST neuron activation increases shunting inhibition to block these inputs. These alternatives and shortcomings of the model may be discussed further.

Reviewer #3:

Remarks to the Author:

In this manuscript the authors aim to describe the neuronal mechanism through which STN-DBS alleviates motor symptoms in PD with a combination of recording techniques (juxtacellular and intracellular

recordings). First, they characterize the changes in activity of motor cortex (M1) neurons in dopamine depleted animals (increase firing rate of pyramidal neurons of 6-OHDA treated rats and mice). Second, they characterize how STN-DBS affects these cortical changes, finding that STN-DBS normalizes the high firing of pyramidal cells in dopamine depleted animals. They hypothesize that local cortical GABAergic network with monosynaptic connection to produce this inhibition. Third, using opto-tagging to identify cortical inhibitory interneurons, they discovered STN DBS decreased the firing rate of PV neurons while it increased firing rate of SST cells. They directly stimulated SST interneurons with optogenetics and find behavioral correction of spontaneous rotations and cross-maze bias. Last, they describe a neural network model to explore how each cell type responds to STN-DBS and direct stimulation of SST.

These scientific issues are interesting and potentially therapeutically applicable providing a scientific basis for cortical modulation as a therapeutic approach, although how one would modulate specific cell type in human without invasive methods is not clear at this point. The involvement of cortical interneurons in this process is novel and interesting. Network modeling provides interesting insights. However, but there are major gaps to infer causal relationships for their hypothesized mechanisms.

While this manuscript is addressing how STN-DBS is beneficial, the behavioral tests are only carried out with optogenetic stimulation of interneurons. Thus, no comparisons can be made between the efficacy/similarities of STN-DBS and modulating interneuron activity. Additionally, there is an effect of PV opto-stimulation, although it is not as robust (Extended data Fig. 4). Given the lack of improvement in hypoactivity in the open field by SST stimulation, (compare figure 3b and S4c), it would be instructive to know whether STN-DBS in their mouse model has the same effect, and how this compares with therapeutic doses of L-DOPA. Complementary and perhaps more commonly used behavioral paradigms such cylinder test and stepping would have been more convincing as well.

Functional manipulations that inhibit SST or PV neurons in combination with DBS would provide a firmer complementary evidence for causal relationship to indicate SST interneurons are necessary as well as sufficient for STN-DBS effect. In addition, it will be helpful to specify whether the model tested that blocking the change in activity of PV or SST neurons abrogates the effect of STN-DBS on the change in pyramidal cell firing rate. Although they note that no decrease in pyramidal cell activity was observed in the absence of DBS-induced current on SST interneurons (Fig.4c), it is not clear what condition this represents.

Network modeling was used to probe how DBS may affect cortical network activity taking into account stimulation of pyramidal neurons, PV interneurons, and SST interneurons. Based on the parameters selected, the modeling did recapitulate many of the experimental findings, though there were some inconsistencies. The authors use the model to suggest that the decrease in cortical pyramidal cell activity by STN-DBS or SST activation can "preserve" network processing of artificial inputs, which is impaired in the simulated PD condition and suggest that this is the mechanism by which STN-DBS confers its therapeutic benefit. Though these are theoretical studies, the authors should comment on how these findings could be tested.

Overall, there is no insight as to why SST neuronal effect predominates the STN-DBS effect. Is it due to differential connectivity of SST interneurons to STN efferents? The authors note asymmetrical connectivity of SST interneurons, but further elaboration is necessary.

Fig. 1 – the authors interpret the pyramidal neuron hyperpolarization caused by DBS to be due to GABAergic inhibition from synaptic input. Could the authors comment on whether a similar hyperpolarization of pyramidal neuron may also be seen if the pyramidal cell is made to fire at a high rate (e.g., 130 Hz) for a prolonged period of stimulation (e.g., as may be expected from DBS antidromic current, or antidromic action potential), which may occur cell-intrinsically via slow afterhyperpolarization via opening of calcium/sodium-gated potassium channels? Unlike a GABAergic mechanism, this would predict the hyperpolarization would still be seen in the presence of AMPA/NMDA/GABA blocker.

The authors mention that "in cells that were antidromically activated by STN stimulation," (line 55): what % of pyramidal cells showed antidromic activation?

Fig. 2d - The authors should comment on why after DBS stimulation has been turned on, there seems to be a time-lag of 1 time-bin (10 s) for pyramidal neurons, and time-lag of 1-2 time-bins (10-20 s) for Sst

neurons, before firing rates changed. And the possible long-lasting effect of DBS for 1-2 time-bins after it's switched off? Is the same time-lags observed in their simulation model?

Fig. 2e – criterion for classifying as excited vs inhibited vs non-modulated vs silent is not specified.

The authors should explain the choice in stimulation parameters for opto-tagging. 100ms is a long time (and it is clear that the PV and SST neurons don't maintain an increased firing rate during the stimulation duration) as well as the intensity (reported as 10mW). These parameters can have long lasting effects on the cells, even though the authors report that they waited for the neurons to "recover".

Explanation as to why the 67Hz was chosen for optogenetic stimulation for behavioral assays should be provided. It will be important to note why the "typical" clinical frequency was not used or provide a discussion as to why other frequencies were used. Along the same lines, can the M1 interneurons in each of these experiment actually follow the stimulation frequency applied?

For the cross-maze experiments: were the two sessions described (stim, no stim) balanced as to which one went first/second? There is a concern for learning with the increased number of trials. Additionally, these sessions could have been combined and have trials stim/no stim randomly until the 30 turn criteria is met.

It would be helpful to the reader to explain in the figures which animals (rats vs mice) were used in the experiments presented (ie with the experimental set-up image).

It is unclear what cells are included in the "cortical cells" that decreased firing rate with STN stimulation (line 47). Is it just pyramidal cells or all cells?

Fig. 4c – why is DBS of PV (vs. Sst) less effective in reducing pyramidal firing here, but seems to be more effective in Supplementary Fig.5C?

Supplementary Fig. 6a – why does the simulation model predict that opto-stimulation of Sst leads to a high asymptotic Sst firing rate (~40 Hz) while opto-stimulation of PV leads to lower asymptotic PV firing rate (~20 Hz), when the opposite seem true with real experimental data shown in Supplementary Fig. 7a? Does this have downstream consequences for the simulation model? (E.g., maybe having PV firing to much in simulation causes pyramidal cells to be too inhibited? If yes, how to reconcile?)

It will be helpful to note how the coordinates for STN was chosen. This structure has a particular organization also receiving input from the PFC. Similarly, how were the coordinates for M1 chosen? The location of all recordings and stimulation seems to be of the anterior portion.

Please correct/align the labels in Figure 2 correctly (bottom of panel b and c).

Line 85: should say juxtacellularly labeled.

The exclusion criteria was defined at "spontaneous ipsilateral rotations". How long was the testing session? Typically the number of amphetamine rotations per min are reported and an a priori minimum is set.

What do the authors consider an opto-response latency? This should be reported in the methods.

Though the authors state in the statistics methods that unless otherwise reported results are displayed as mean±SEM, it would be helpful to the reader to have it stated in the individual figures.

Reviewers' comments:

Reviewer #1 (Remarks to the Author):

Paper by Vandecasteele and coworkers investigated the cortical effect of subthalamic nucleus-DBS in animal models of Parkinson's disease. Authors found that DBS rescued the pathological hyperactivity of motor cortex pyramidal cells. Specifically, authors find a selective involvement of somatostatin interneurons (SST). Finally by means of an optogenetics approach, authors demonstrated a beneficial effect of cortical modulation in alleviating motor symptoms in model of Parkinson's disease. Paper is clear and well written and technically advanced. Statistical and computing approach is accurate.

The following point require attention.

- Protocol for stimulation of Chr2 in optogenetics experiments has been reported as 3ms 67 Hz. This frequency of stimulation seems to fast for ChR2 kinetic. Did the authors test it by an in-vitro input-output curve? Use of this protocol should be supported by references or discussed it.

As Reviewer #1 points out, the 67 Hz protocol is slightly faster than the maximal frequency that ChR2-H134R is reported to be able to follow faithfully (Lin et al., 2009 PMID 19254539; Berndt et al., 2011 PMID 21504945) - though it is notable that (i) these tests were done in cells (neuronal cultures and pyramidal cells, respectively), prone to have slower kinetics than the here targeted interneurons, and (ii) both reports show that ChR2-H134R is better able to follow higher frequencies with increasing light intensities, which is the case in our study.

We chose the 67 Hz frequency knowingly, as we did not want all opto-stimulated interneurons in the vicinity of the optic fiber to be activated synchronously at each light pulse. Indeed, since synchronous oscillations are a hallmark of Parkinson's disease (Singh, 2018, PMID 29381817), we wanted to avoid imposing a strong rhythm to the whole cortical networks. Therefore, we chose a frequency higher than 50 Hz (20 ms period), which PV and SST interneurons would be able to follow. Rather, our goal was that each pulse elicits the activity of a proportion of neurons stochastically picked among the population. We also wanted to avoid choosing a frequency so high that ChR2-H134R would not have time to close, which could lead to the silencing of nearly all neurons with the continuous application of the protocol. We therefore were capped by the tau-off of ChR2: ~15-20 ms (Lin et al., 2009 PMID 19254539); we chose the lowest value of this range, which corresponds to 67 Hz (15 ms period).

To validate this choice (67 Hz), we recorded opto-identified PV and SST interneurons during 1-minute continuous stimulation at 13 Hz (a frequency that ChR2-H134R should be able to follow easily), 67 Hz (our chosen frequency) and 130 Hz (the electrical DBS frequency) (new Supplementary Figure 4a and 4b). Among neurons that were well entrained by 3ms pulses, we observed that at 13 Hz, all PV and SST neurons were able to fire at every light pulse. At 130 Hz, the proportion of responding cells at a given pulse decreased to 7% of the SST and 10% of the PV interneuron (measured during the last 10s of 130 Hz-stimulation), with 45% of the SST and 18% of the PV being nearly or totally silenced (0-1 % success rate per pulse). Opto-stimulating at 67 Hz was able to maintain a mean of 26% of SST and 55% of PV cells responding at a given light pulse during the last 10s of opto-stimulation. Furthermore, the responding cells were not the same at each light pulse, with 82% of the SST and 100% of the PV interneurons participating with a success rate per pulse > 1%. Therefore, stimulating at 67 Hz appeared as a good compromise, allowing a neither negligible (130 Hz) nor overwhelming (13 Hz) proportion of interneurons to fire.

To answer this point, new analysis and additional experiments have been performed, and are now illustrated in the new Supplementary Figure 4a and 4b, and detailed in the Results and Methods sections.

“We chose a stimulation frequency (67Hz) that was close to the opsin tau-off (~15-20ms⁶⁴). Indeed, higher frequencies such as 130Hz (the same frequency as electrical DBS) would lose efficiency quickly, while lower frequencies such as 13 Hz, able to entrain all the neurons at each pulse (Fig. S4a and S4b), might impose pathological-like synchronous activation⁶⁵. DBS stimulation: electrical current (60μs, 120μA, 130Hz).”

- Authors report a reduction in firing rate of pyramidal neurons recorded with intracellular technique. Are the authors measured spontaneous firing activity? All recorded neurons were spontaneously active?

We indeed observed that DBS decreased the firing rate of pyramidal cells recorded with intracellular technique; this result was observed in both the spontaneous activity (new Fig 2b, previously in Fig1) and the evoked activity (new Fig 2c, previously in Fig1).

The legend of the Figure 2 and the corresponding Results have been modified to clarify that Figures 2b and 2c illustrate spontaneous and evoked activity, respectively.

Among the neurons recorded with intracellular technique, 95% were spontaneously active (n=19/20). The neuron which was not spontaneously active is also included in Figure 2b. This information is now in the Figure 2 Legend.

- Figure 1b: is not clear how frequency is represented in this image.

The legend of Figure 1b has been modified to clarify the nature of the representative raster plots.

Moreover, we now provide a new Figure (Fig. 1) illustrating exclusively the extracellular recordings (intracellular recordings are illustrated in the new Fig. 2) with new analysis included: heatmaps and averaged time courses of DBS effect on spontaneous cortical activity, including the activity after the offset of DBS (new panels in Fig. 1b and 1c). This new representation (heatmap) shows frequency along time (before, during and after DBS) of individual neurons (new Fig. 1c and Fig. S1b). In addition, we have distinguished in the average time courses the activated and inhibited pyramidal cells.

These three redesigned figures (Fig. 1, 2 and S1) allow for better clarity and we now provide more informative legends.

- Figure 1f is not particularly clear. A better representation should be provided.

This panel (new Fig. 2e) has been redesigned. We added a representation of the current steps applied to the pyramidal cells, selected 11 representative current traces (instead of 15) for clarity, and extracted the inset (reversal potential) as a full panel. The associated legend is now more detailed.

In addition, we added a zoom of the PSP evoked by STN single stimulation (for -1.4 nA injected current); which also more clearly shows where the PSP amplitude was measured (early phase of the PSPs).

The intracellular recordings are now grouped in a separate figure (new Fig. 2) apart from the extracellular recordings (new Fig. 1).

- Does the STN stimulation affect other cellular properties of pyramidal neurons, as I_h current or AHP current?

In the revised version of our manuscript, we now show the effect of DBS on the membrane time constant, input resistance, I_h current and the AP threshold. This is detailed in the Results:

“We further confirmed that DBS decreased the spontaneous firing rate ($p < 0.001$, $n = 20$) of M1 pyramidal neurons (Fig. 2b). This decrease was accompanied by a hyperpolarization of -3.7 ± 1.0 mV of their membrane potential ($p < 0.001$, $n = 20$) (Fig. 2b), and a decrease in their membrane time constant ($p = 0.0390$, $n = 18$), input resistance ($p = 0.0497$, $n = 19$) and AP threshold ($p = 0.0263$, $n = 20$), without affecting their I_h current ($p = 0.2336$, $n = 17$).”

These results suggested that pyramidal cells are less excitable under DBS than without DBS, and this could participate to decrease the firing rate of pyramidal cells during DBS.

- Effect of DBS stimulation onto cortical firing activity was transitory and or reversible?

To answer this question, we performed new analysis. In the new Figure 1 (panel c) and new Supplementary Figure 1 (panel b), we added the heatmaps and averaged time courses of DBS effect on cortical activity, including the activity after the offset of DBS. The normalized activity under DBS fell below 2xSD of the baseline during the whole DBS duration, indicating a stable effect of DBS. At the offset, the normalized activity is partially reversible.

These results (in rats) are consistent with the time course observed in identified PV, SST and pyramidal cells recorded in mice (new Fig. 3d and 3e).

In mice, cell identity was assessed by opto-identification and PCA clustering, allowing to classify PV, SST and pyramidal cells. In rats the identification was achieved solely with waveform characteristics. Accordingly, we are now addressing the question of the stability and reversibility of DBS effects in Figure 3, and in the corresponding Results section:

“The activity of pyramidal, PV and SST cells was stable under DBS, and reversible (with a shorter delay after the DBS offset for PV cells than for SST and pyramidal neurons).”

- DBS reduce firing activity of PV interneurons but increase firing activity of SST interneurons. According with the normalized firing activity the overall GABAergic input onto pyramidal neurons seems to be quite unaltered. Have the authors any data about the synaptic inhibitory input onto pyramidal cells? Are the interneurons activity downregulated in parkinsonian models? In vitro experiments could help to evaluate this point.

According to Reviewer#1's comment, we performed additional experiments using *in vivo* juxtacellular recordings (new Fig. 4a-b). In addition, we chose to perform *in vivo* patch-clamp (whole-cell) recordings (new Fig. 4c-f), instead of *in vitro* as proposed by Reviewer#1, to have as much as possible the neuronal network preserved and be in similar conditions than the others experiments conducted in this manuscript.

To test whether the spontaneous firing activity of M1 interneurons is altered in 6-OHDA-lesioned mice, we performed juxtacellular recordings with opto-identification and PCA clustering of PV and SST cells in sham (PV cells=27 and SST cells=26) and 6-OHDA-lesioned (PV cells=14 and SST cells=15) *Pv::Chr2* and *Sst::Chr2* mice (Fig. 5a). We found that 6-OHDA lesions did not affect the firing rate of neither PV nor SST cells compared to sham (new Fig. 4b). These results suggest that the pyramidal hyperactivity in parkinsonian models is not due to changes in electrophysiological activity of cortical GABAergic populations.

These data are detailed in a new chapter, in the Results section:

« *In vivo firing rates of M1 PV and SST interneurons are similar in sham and 6-OHDA-lesioned mice*

“Since DBS both normalizes pyramidal cell hyperactivity and differentially recruits PV and SST interneurons, we further explored the firing rate modulation of interneurons in the parkinsonian condition. Indeed, hyperactivity of pyramidal cells observed in 6-OHDA-lesioned animals could be caused by a decreased activity of GABAergic interneurons. We performed in vivo juxtacellular recordings of neurons opto-identified and clustered by

principal component analysis as PV or SST cells in anesthetized sham (PV cells=27 and SST cells=26) and 6-OHDA-lesioned (PV cells=14 and SST cells=15) Pv::ChR2 and Sst::ChR2 mice (Fig. 4a-b). The spontaneous firing activity of PV cells was similar in sham and parkinsonian Pv::ChR2 mice ($p=0.8391$ with $n=32$ in sham and $n=14$ in 6-OHDA-lesioned mice). Similarly, spontaneous firing rate of SST cells was not different in sham and 6-OHDA-lesioned mice ($p=0.2112$ with $n=32$ in sham and $n=15$ in 6-OHDA-lesioned mice). These results suggest that the pyramidal hyperactivity in parkinsonian models is not due to changes in electrophysiological activity of cortical GABAergic populations.”

Next, to investigate synaptic inhibitory inputs onto pyramidal cells, we performed *in vivo* whole-cell recordings in Pv::ChR2 and Sst::ChR2 anesthetized mice (new Fig. 4c-f). We recorded 7 and 9 pyramidal cells subjected to opto-activation of PV and SST interneurons, respectively. We performed either single stimulation (with 4 different durations: 3, 5, 10 and 20 ms) or at 67 Hz for 5 seconds (3 ms pulses). These opto-activations were applied for three membrane potential ranges (-95: -80), (-65: -50) and (-40: -30) mV, corresponding to -100, 0 and +100 pA injected currents.

These results are detailed in a new chapter:

“Characterization of IPSP at pyramidal cells evoked by opto-activation of PV or SST cells.

To investigate the synaptic weight of the inhibitory inputs onto pyramidal cells, we performed in vivo patch-clamp whole-cell recordings in anesthetized Pv::ChR2 and Sst::ChR2 mice (Fig. 4c-f). We recorded pyramidal cells upon opto-activation of PV or SST interneurons. First, we characterized the evoked-PSP following single opto-stimulation (3-20 ms). The opto-activation of PV and SST cells evoked PSPs of similar amplitudes in pyramidal cells, regardless of the membrane potential ($p=0.3397$, $n=8$ pyramidal cells in Pv::ChR2 vs. 6 in Sst::ChR2 held at -95/-80 mV; $p=0.8632$, $n=7$ vs. 7 held at -65/-50 mV and $p=0.8838$, $n=7$ vs. 9 held at -40/-30 mV) (Fig. 4c1 and 4c2). We analyzed the voltage-dependency of opto-PSPs, which reversed at -71.5 mV in Pv::ChR2 ($n=6$ pyramidal cells) and -75.2 mV in Sst::ChR2 ($n=4$) mice ($p=0.0542$) (Fig. 4c3), i.e. close to the calculated chloride reversal, -71.9 mV. Single pulse opto-activation of PV ($n=7$) and SST ($n=6$) cells induced PSPs whose area, amplitude, rise time and delay to peak increased with increasing opto-pulse duration (Fig. 4d1-d4, 2-way repeated-measures ANOVA). There was no difference between PSPs evoked by PV or SST opto-activation, except for the delay to peak amplitude which was shorter upon PV than SST opto-activation ($F_{1,11}=66.39$, $p<0.0001$). We then investigated the effects of opto-activation of PV or SST cells at 67 Hz for 5 seconds (see Methods and Fig. S4a-b for opto-stimulation frequency selection) (Fig. 4e and 4f). We observed that opto-activation of PV or SST cells induced a hyperpolarization of similar peak amplitude ($p=0.2270$, Fig. 4f1), with a shorter delay upon PV activation ($p=0.0124$; Fig. 4f2) that strongly inhibited the firing activity of pyramidal cells ($p=0.0392$, $n=5$ in Pv::ChR2 mice, $p=0.001$, $n=8$ in Sst::ChR2 mice) (Fig. 4e2). However, in the latter part of the pulse, opto-activation of SST cells induced a less pronounced membrane hyperpolarization than opto-activation of PV cells ($p=0.0371$; Fig. 4f3), together with a partial release of the spiking inhibition only in Sst::ChR2 mice ($p=0.0326$; Fig. 4e). In addition, the variance of the membrane potential was larger under opto-activation of SST than PV cells ($p=0.0371$; Fig. 4f4), denoting a strong shunting of synaptic activity reaching the soma by PV opto-activation, whereas some synaptic activity subsisted (leading eventually to spikes) under SST opto-activation.”

All these data are illustrated in the new Figure 4.

- Authors should provide details about reversal potential of the Cl calculated for the intracellular solution used to perform experiments.

We now provide information about the estimation of the reversal potential of the chloride in the intracellular recording experiments, in the Results and Methods sections. Based on previous *in vivo* intracellular recording studies performed in the lab (Dégenétais et al., 2003, Cerebral Cortex) and by other teams (*in vivo*: Agmon and Connors, 1992; Baranyi et al., 1993; Nunez et al., 1993; in neocortical slices: Avoli, 1986; Connors et al., 1988; Hablitz and Sutor, 1990; Higashi et al., 1991; Cox et al., 1992), we proceeded as follow: we recorded pyramidal cells with 2 M potassium acetate in the intracellular solution (pipette displayed a high impedance: 40–70MΩ). Upon STN stimulation and different holding membrane potential, we observed that the early PSP amplitude canceled around -70 mV. In a previous study (performed in our Institute), “when the recording microelectrode was filled with potassium chloride instead of potassium acetate, the early IPSP was depolarizing and its reversal potential was shifted to a less negative value” (Dégenétais et al., 2003); the late phase of the PSP hardly reversed, its amplitude reached zero at -77mV and this value was not modified by the replacement of potassium acetate by potassium chloride in the microelectrode (Dégenétais et al., 2003), and this late phase was most likely due to potassium currents gated by GABA_B receptors.

These data are consistent with previous studies performed in the neocortical slices indicating that evoked inhibitory responses were composed of an early IPSP largely resulting from a Cl⁻ current through GABA_A receptors and of a late IPSP due to K⁺ currents gated by GABA_B receptors (Avoli, 1986; Connors et al., 1988; Hablitz and Sutor, 1990; Higashi et al., 1991; Cox et al., 1992).

In the new Figure 2e: an inset has been added to show the early and late phases of PSP and where the measurements for the “peak amplitude” were taken to determine the chloride reversal.

- PV interneurons stimulation by opto-activation only partially reproduce the improvement in motor symptoms induced by SST opto-stimulation. How authors justify this results? A simple higher inhibition seem not enough to rescue motor symptoms and potential differences in strength of synaptic input could be supposed.

Indeed, PV opto-activation only improves rotational bias, but not the asymmetry observed in the cross-maze task, while SST opto-activation improves both symptoms. Furthermore, we added a new behavioral task (cylinder task, see Fig 5c1-c6), to test the asymmetry in front paw use, which confirmed these results: SST opto-activation, but not PV opto-activation, improves front paw bias. Interestingly, our modeling results are consistent with the experimental findings: SST opto-activation is more efficient than PV opto-activation for restoring cortical information processing capabilities (Fig. 7). We were able to test several hypotheses that could explain these results:

(1) The simplest explanation would be that the SST have a stronger synaptic strength onto pyramidal neurons. However this is inconsistent with previous studies (Pfeffer et al., 2013).

(2) Alternatively, our opto-stimulation conditions might be more efficient in recruiting SST rather than PV cells, stemming for example from a different distribution of their dendritic tree across cortical layers, inducing a stronger "collective" synaptic strength of SST in our conditions. We tested this hypothesis by performing new experiments using *in vivo* whole-cell recordings of cortical pyramidal cells in *Pv::ChR2* and *Sst::ChR2* mice (Fig. 4c-f), and observed that the SST opto-activation was not more efficient than PV opto-activation for single opto-stimulation pulses (Fig. 4c) in our conditions.

(3) The higher impact of SST on behavior and processing capabilities could also result from a more efficient recruitment of SST interneurons in the case of continuous 67 Hz stimulation. We tested this hypothesis using *in vivo* juxtacellular recordings of opto-identified PV and SST interneurons (Fig. S4a) opto-stimulated for 1 min at 13, 67 and 130 Hz, and consistently observed the opposite result, i.e. that PV cells were more faithfully entrained than SST cells.

(4) Another possibility could stem from differences in short term plasticity properties of PV and SST synapses onto pyramidal cells, since SST-mediated synaptic inhibition depresses only slightly with repeated activation, in contrast to the rapid depression undergone by IPSPs from PV synapses (Cardin et al., 2018) and excitatory inputs onto SST cells are also strongly facilitating (Urban-Ciecko and Barth, 2016). The stronger effect of SST would then be revealed along a continuous stimulation. We tested this hypothesis using *in vivo* whole-cell patch-clamp recordings (Fig. 4c-f), and *in vivo* juxtacellular recordings of cortical pyramidal cells (Fig. S4b) under 67 Hz continuous opto-stimulation in *Pv::ChR2* and *Sst::ChR2* mice. Both techniques, as well as our modeling results (Fig. 6d and Fig. S7), show that at 67 Hz, continuous PV opto-activation is actually more efficient than SST opto-activation at inhibiting pyramidal cells. As the reviewer suggests, this proves that a higher inhibition is not enough to rescue motor symptoms. Actually, these results raise the question of whether the inhibition provided by PV interneurons is too efficient, which counteracts potential beneficial effects on behavior or cortical processing. However, if this was the only reason explaining the difference in PV and SST opto-activation outcome, we would expect an adverse effect of PV activation in control animals. Indeed, if PV activation leads to an inhibition that is deleterious to cortical processing, their unilateral activation on a normal, symmetrical brain, would lead to ipsilateral rotations. We did not observe an effect of either PV or SST opto-activation in control mice (Fig. S4), suggesting that the strong inhibition provided by PV opto-activation is not deleterious in itself. In addition, in the model, we also tested the information capacities of the network under lower amplitude stimulation of PV interneurons, thus avoiding a strong silencing of pyramidal cells, and still found weak correlation coefficients between network responses and the stimuli, compared to SST opto-activation (Fig. S6).

(5) From a pathophysiological point of view, SST activation could be more efficient than PV activation if it compensated a pathological imbalance, such as a hypoactivity of SST interneurons in parkinsonian conditions. We tested this hypothesis using juxtacellular recordings of opto-identified PV and SST cells in control and 6-OHDA mice (Fig. 4b), and did not observe a change in the spontaneous activity of either population. However, it should be noted that these results are obtained under anesthesia, and would need to be confirmed in awake conditions.

(6) The connectivity between PV and SST interneurons and pyramidal cells likely plays a role in the different outcome of their activation. Indeed, in L5 motor cortex, PV neurons are involved in strong feedback loops with pyramidal neurons, while SST neurons, in part recruited by L2/3 pyramidal neurons, provide disynaptic feedforward inhibition (Apicella et al., 2012). Our model results show that this asymmetrical connectivity is crucial for DBS effect on pyramidal cells (Fig. 6c), and has important consequences on the temporal structuring of the network activity patterns (Fig. 7b-c and Fig. S6).

(7) Lastly, an interesting hypothesis stems from the different position of PV and SST synapses on pyramidal cells. Indeed, the position of SST terminals onto pyramidal dendritic branches, leading to a local and direct inhibition of excitatory inputs, may explain why SST activation allows for better information processing, compared to PV somatic-targeted inhibition (Urban-Ciecko and Barth, 2016; Ramaswamy et al., 2017).

Overall, SST interneurons could thus be more efficient than PV cells at exerting a sustainable inhibitory influence by filtering excitatory inputs onto pyramidal neurons, rather than simply silencing pyramidal cell output, resulting in an increased performance of motor function. A more detailed model, including a multi-compartmental description of pyramidal cells as well as short-term plasticity properties, would allow exploring how these differences in PV and SST physiology could account for the differences in behavioral and processing outcome of their opto-activation.

These points were added (in a condensed version) at several places in the corresponding Results (new experiments) and Discussion parts, and illustrated in the new Figures 4, S4, 6, 7, S6 and S7.

- Paper shed light to an interesting topic, however the overall interpretation seems oversimplified: Although cortical implications seem clear, authors do not consider others potential effect of DBS onto others basal ganglia structure (STN globus pallidus). A better consideration of this aspect would improve the paper.

We fully agree with Reviewer#1. Indeed, while this study focuses on cortical effects of DBS, we do not mean to exclude its effects on basal ganglia nuclei. Actually, we consider that the cortical effects we observed correspond to the sum of the direct and indirect (through basal ganglia loops) effects of DBS. The DBS-mediated currents used in the model represent the sum of these effects. This is now clearly mentioned in the Results:

" Finally, DBS-mediated currents were modeled as instantaneous excitatory inputs to all cells, with no preconception of the pathway involved to reach each population (directly via antidromic or orthodromic connections from STN to cortex, or indirectly through basal ganglia loops), thus representing the sum of DBS network effects. "

The involvement of basal ganglia in cortical effects of DBS is also now mentioned in the Discussion:

"Concurrently, cortical effects of DBS could occur via recruitment of other basal ganglia-thalamo-cortical loops." [] "The concurrent recruitment of these anatomical pathways likely results in a new cortico-basal-ganglia wide state of equilibrium. This DBS-induced state stabilizes over time, which would explain the observed onset and offset delays in pyramidal and SST responses to DBS."

Furthermore, basal ganglia modulation of DBS can have behavioral effects independently of cortical effects. We now include new behavioral experiments, testing the effects of DBS in parkinsonian mice (new Fig 5b5-d5 and Fig S4c). In contrast with our cortical stimulation, DBS in mice improves hypolocomotion. This suggests that the cortical stimulation does not fully recapitulate DBS effects, therefore that DBS effects on basal ganglia structures contribute to the beneficial locomotor effects. This is now mentioned in the Discussion :

" Yet the absence of effect following opto-activation on hypolocomotion suggests that non-cortical effects of DBS (i.e. direct modulation of basal ganglia structures⁴⁻⁶), which are unlikely to be mimicked as efficiently by our cortical intervention, also contribute to the beneficial locomotor effects."

Reviewer #2 (Remarks to the Author):

The manuscript from Vandecasteele et al. describes a series of experiments that characterize how the cortical microcircuit in M1 responds to deep brain stimulation of the subthalamic nucleus (STN-DBS). The main finding is that STN-DBS is associated with elevated activity of SST interneuron in M1, which the authors show is a contributing factor to the therapeutic effect of STN-DBS in rodents. There are several strengths for this paper. The authors have presented high-quality data. The writing is crisp and clear. The experiments make sense, and the proper controls have been performed. As someone who is not familiar with the STN-DBS or Parkinson's disease literature, I cannot speak to the novelty of the findings. However, I believe on the strengths of the data alone – because it is rigorously collected and using contemporary electrophysiological, optogenetic and behavioral techniques in rodents – this paper is a strong contribution. I have several comments that may improve the clarity of the presentation. It would be helpful to address the comments, but whether the authors address them would not affect my overall opinion of the paper.

Minor comments:

- It would helpful to know from the main text for experiments associated with Figure 1a, whether the in vivo juxtacellular recording was done in anesthetized or awake rat.

We added in the main text and in the Figure 1 legend that *in vivo* juxtacellular recordings were done in anesthetized rats.

Also it is unclear how the authors know the recorded cortical cells are layer 5 pyramidal neurons. Is this based on spike waveform alone or is there some staining with the juxtacellular recording?

Indeed it is important to mention that the neurons were considered as pyramidal neurons from deep layers of M1 based on the recording depth (1422 μ m – 2622 μ m) and on a combination of electrophysiological characteristics. To clarify this point, we added in the Material & Methods part this information:

“Single-unit activity was recorded extracellularly in deep layers of M1 (depth range of recorded neurons: 1422-2622 μ m) using glass micropipettes (10–15M Ω) filled with NaCl (1 M) and identified as pyramidal based on their waveform features (with waveform width from depolarization to trough >1ms; Fig. S1a).”

This is also now illustrated in the new Supplementary Figure 1 (panel a).

- Figure 1b: Should show data points for individuals in addition to summary bar graphs.

We added in the revised manuscript the individual data in the new Figure 1 and in the related Supplementary Figure 1.

In addition, we are now providing heatmaps showing the firing activity of the individual neurons before, during and after DBS in 6-OHDA-lesioned rats (new Fig. 1c) and in sham rats (new Fig. S1b).

- Figure 1c and 1d: The main text and figure could be more clear in terms of when the DBS was applied – when did it start and for how long relative to the recording?

In the new Figure 2b and 2c (ex Fig. 1c and 1d), we added horizontal bars to indicate precisely when DBS occurred.

In these experiments, spontaneous firing of pyramidal cells were recorded for at least 150 seconds before and during DBS. Then, depolarizing current steps were applied to study the evoked activity (I/F curve) without DBS and with DBS for 100 seconds.

This information has been added in the Results section (and in the Figure 2 legend):

“Spontaneous firing of pyramidal cells were determined before and during 150 seconds of DBS.”

“To characterize active membrane properties of pyramidal cells with and without DBS, we applied current steps before and during 100 seconds of DBS.”

- The electrophysiology was done in anesthetized rats, whereas behavior is done with awake mice. As the authors stated, prior studies have noted that anesthesia substantially influence the activity patterns of SST interneurons. There should be some discussion on this caveat when trying to use the electrophysiology to draw inference about the behavioral results, or vice versa. We fully agree with Referee #2 for this comment; although to be precise, we also performed electrophysiology in anaesthetized mice. To answer this point, as suggested by Reviewer#2, we added in the Discussion section:

“The similar firing rate of interneurons observed in (anesthetized) sham and 6-OHDA animals suggests that this therapeutic effect does not stem from a pathological hypoactivity of interneurons in parkinsonian conditions. However, the dampening effect of anesthesia on SST firing patterns could occlude differences between sham and lesioned animals. Similarly, our data pointing to SST recruitment by DBS was obtained in anaesthetized rodents. While we can reasonably hypothesize that in the absence of the dampening effect of anesthesia, DBS-activation of SST interneurons would be even higher, and thus exert an even stronger inhibition onto the pathological hyperactivity of pyramidal neurons, this remains to be confirmed in future experiments.”

- The authors showed that STN stimulation increases the activity of SST interneurons. Is this due to axons from STN neurons to SST interneurons? Or is the activation due to DBS recruiting neighboring regions or axons passing by STN? Even if there is no data, some discussion of the possibilities would be helpful.

We are now discussing this point:

“Several pathways could account for SST recruitment by DBS: antidromic activation of the cortico-subthalamic fibers, orthodromic activation of subthalamo-cortical fibers, and/or basal ganglia-thalamo-cortical loops^{4,11}. The activation of SST interneurons by antidromic axonal reflex is unlikely to occur since cortico-STN pyramidal cells lack collaterals to SST interneurons²⁴. The orthodromic STN-cortex pathway could recruit SST neurons, since it mainly targets superficial cortical layers³⁴, where pyramidal cells directly connect to L5 SST interneurons³⁵. Concurrently, cortical effects of DBS could occur via recruitment of other basal ganglia-thalamo-cortical loops. Other M1 interneurons than PV and SST might be at play during DBS, such as VIP cells³⁶, since their inhibition would result in the disinhibition of SST cells. The concurrent recruitment of these anatomical pathways likely results in a new cortico-basal-ganglia wide state of equilibrium. This DBS-induced state stabilizes over time, which would explain the observed onset and offset delays in pyramidal and SST responses to DBS.”

- Figure 2d: A few features are interesting in this PSTH: that the recruitment of SST activity increases over time during DBS, and that the SST activity remains heightened for another ~30 s after DBS ended. Any idea why this is the case?

Indeed, these onset and offset delays observed in both SST and pyramidal populations suggest that the effects of DBS involve both direct (subthalamo-cortical) and indirect (basal ganglia induced) modulations of cortical cells which leads to a DBS induced new state of equilibrium. This observation is now noted in the Results, and discussed in the Discussion as follows:

Results:

“Interestingly, the kinetics of DBS effects on pyramidal cells mirrored those of SST interneurons: we observed similar delays between the onset of DBS and its effects on pyramidal and SST cells, and also at the DBS offset (Fig. 3e). Consistently, the modulation of pyramidal cell was strongly correlated with SST interneuron activity during and after DBS ($r=-0.91$, $p<0.001$).”

Discussion:

“The concurrent recruitment of these anatomical pathways likely results in a new cortico-basal-ganglia wide state of equilibrium. This DBS-induced state stabilizes over time, which would explain the observed onset and offset delays in pyramidal and SST responses to DBS.”

- For Figure 3, the PV interneuron activation is an important comparison. The authors have done this experiment but the results are buried in Extended Data Fig. 4. Either outcome for the PV manipulation experiments - whether that help to restore or has no effect on behavior – is interesting. PV neurons were studied along with SST cells at all stages of the studies, from electrophysiology to behavior to modeling, so it is odd to leave it out of the main figures for behavior. The results should be part of the main figures. To a lesser degree, the results from stimulation of SST interneurons on sham mice are also interesting, and should also be moved to be the main figures. I know they don't add nicely to the 'story', but they are intriguing observations that could lead to future studies.

In the revised version of our manuscript, we moved all results regarding opto-activation of PV cells in *Pv::ChR2* 6-OHDA-lesioned mice in the new main Figure 5 (panels b3-d3). Interestingly, these results show that opto-activation only partially reproduced the improvement in motor symptoms induced by SST cell opto-activation.

In addition to the open field and cross-maze tasks, we have added the cylinder test as an additional task to evaluate the asymmetrical locomotor behavior. We are now presenting new results showing that opto-activation of SST, but not PV cells, allows to decrease the asymmetry in front paw usage preference (new Fig. 5c2 and 5c3).

We agree with Reviewer#2 that results from opto-activation on sham mice are interesting. We chose to leave these experiments in Supplementary Figure 4, because in the new main Figure 5 we added 3 new conditions (opto-activation of PV cells in *Pv::ChR2* 6-OHDA-lesioned mice, DBS in wild-type 6-OHDA-lesioned mice and L-Dopa-treated wild-type 6-OHDA-lesioned mice) and one task (the cylinder test), making the new figure quite crowded. Nevertheless, experiments performed in sham mice are now more extensively commented in the Results section:

*“Interestingly, opto-activation of cortical interneurons in *Sst::ChR2* or *Pv::ChR2* sham-mice did not induce an asymmetrical behavior contralateral to the opto-activation neither in the open field (Fig. S4d) nor in the cross-maze task (Fig. S4e). This indicates that the reduced asymmetry observed upon SST opto-activation in 6-OHDA-lesioned mice is due to an improvement of pathological symptoms rather than a generic effect of unilateral opto-activation of cortical interneurons. »*

Following Reviewer#2 comments on the experimental side, regarding the computational model, we have also included data of the opto-activation of PV cells in the main figures illustrating modeling (new Fig. 6 and 7).

- For the model, the analysis of injecting various waveforms, and then comparing the fidelity of the representation is quite innovative and appropriate. I have, however, a few questions about the construction of the in silico circuit. Why is the STN-DBS modeled as an excitatory

input to all cell types – what justify this? Perhaps more importantly, do the connection strengths and probabilities in the simulation resemble physiological values? With so many connections, it seems that a fit that match the observed STN-DBS effects is going to be possible, but it is more a matter of whether the fitted circuit is even close to physiology.

“Why is the STN-DBS modeled as an excitatory input to all cell types – what justify this?”

According Reviewer#2 comment, we have added to the revised manuscript a justification for the choice of STN-DBS interaction as an excitation to all cell types. Essentially, we assumed that STN-DBS could be modeled as an excitatory input to all cell types for two main reasons: -Pyramidal cells can be antidromically activated through the hyperdirect pathway (Baker et al., 2002; Degos et al., 2013). This can also lead to PV interneurons activation through pyramidal cells axon collaterals (antidromic axonal reflex). In addition, pyramidal cells can be orthodromically activated (from thalamo-cortical reafferent loop) (Degos et al., 2008). Moreover, Figure S5c explores how changes in DBS input strengths modify the dampening of pyramidal cell activity, and shows that the model does not sensitively rely on the choice of input to PV cells, and persist even in the case of slightly inhibitory input to PV neurons.

- Since we did not explore experimentally the precise mechanisms leading to SST neurons activation under DBS, the model architecture remained focused on only three populations (pyramidal, PV and SST cells) of L5 motor cortex. The choice of connectivity weights was also based on experimental evidence (see below*). In particular SST neurons are known to receive very little excitation from L5 pyramidal cells (Apicella et al., 2012). Thus the only way for SST neurons to become activated, as observed experimentally, was to consider that DBS ultimately triggers an excitatory input in SST neurons. This activation could originate from different pathways that were not included in the model:

We thus added in the results: *“Finally, DBS-mediated currents were modeled as instantaneous excitatory inputs to all cells, with no preconception of the pathway involved to reach each population (directly via antidromic or orthodromic connections from STN to cortex, or indirectly through basal ganglia loops), thus representing the sum of DBS network effects.”*

Yet, since we cannot estimate precisely the overall excitatory inputs and their relative contribution to each neuronal population of the network, we also chose to vary the strength of the STN-DBS input to each of the populations. The corresponding results are emphasized in the heatmaps in the Figures 6b and S5c.

“Perhaps more importantly, do the connection strengths and probabilities in the simulation resemble physiological values? With so many connections, it seems that a fit that match the observed STN-DBS effects is going to be possible, but it is more a matter of whether the fitted circuit is even close to physiology.”

The Referee is absolutely correct that given the number of parameters, there is no unique choice of network that fit the observations. Our choice of connectivity weights was guided by experimental evidences, and we now underline this in the Results section.

“The choices of intrinsic parameters defining each population (Table S1), network architecture and synaptic strengths (Fig. 6a) were guided by experimental²³⁻²⁹ and modeling³⁰ data. In particular, we modeled PV and SST interneurons as fast-spiking and low-threshold spiking neurons, respectively. As for the network architecture, we set lower conductances for excitatory synapses compared to inhibitory ones^{31,32}. Both PV and SST interneurons inhibited pyramidal cells, with similar connection probability but a stronger synaptic weight from PV to pyramidal cells^{25,26}. Importantly, only PV cells received excitatory feedback²⁴. The choice

of asymmetric inhibition strength between PV and SST neurons, as well as the respectively weak and strong self-inhibition for SST and PV cells is consistent with previous reports^{26,31}.”

(*) More precisely, we used the following studies to infer the relative weights of the connections used in the model:

- Apicella et al. (2012) analyzes intra-laminar excitation to fast-spiking (FS) and low-threshold spiking interneurons (LTS) in motor cortex. They indicate that excitatory inputs to LTS neurons from L5 pyramidal cells are weak and/or sparse, while FS receive more of such inputs. In their last set of experiments consisting of opto-stimulating L5 ChR2-expressing cortico-striatal neurons, they indicate that the size of EPSPs was ~5 times stronger in FS (13.7 mV) than in LTS interneurons (2.5 mV). In the model, we assume that the connectivity profile of FS and LTS neurons map those of PV and SST neurons, since we endow PV and SST with FS and LTS intrinsic properties respectively.

We also show (new Fig 6c) the impact of adding an excitatory connection from pyramidal to SST neurons: increasing the strength of this excitatory connection leads to a progressive extinction of the decrease in pyramidal cells activity under DBS.

- Tanaka et al. (2011) explore the inhibitory strength provided by SST and PV interneurons onto pyramidal cells in L5 motor cortex. While they obtain different comparative results for two subtypes of FS (depending on their arborization) and SST neurons, we considered homogeneous properties among all PV/FS and among all SST cells. Overall, Tanaka et al. (2011) highlight that the connection strength of PV to pyramidal cells is higher relative to SST neurons: this was translated in the model with a 2.8 vs. 2.2 nS synaptic weight from PV vs. SST to pyramidal cells, with similar connection probability (Fig. 6a). This choice is also in agreement with the work from Pfeffer et al. (2013) in the visual cortex, in which only the synaptic strength (and not the connectivity probability) between PV vs. SST onto pyramidal cells varies.

- In addition, the study from Pfeffer et al. (2013) and the reviews from Ramaswamy et al. (2017) and Urban-Ciecko (2017) add that in visual and somatosensory cortices, SST neurons connect to PV neurons while they selectively avoid forming connections with neighboring SST neurons. We extrapolated these findings to the motor cortex, including in the model architecture, very low and weak connectivity ($p = 0.1$, $w = 1.6$ nS) within SST interneurons and a stronger inhibitory effect of SST onto PV ($p = 0.3$, $w = 2.4$) (Fig. 6a). On the other hand, PV neurons have been shown to strongly inhibit each other (which we translated into: $p = 0.6$, $w = 2.8$ nS) while they provide no or little inhibition onto SST neurons ($p = 0.1$, $w = 1.6$ nS) (Fig. 6a). We previously tested the model without the intra-SST and PV->SST connections and observed similar results compared to the full model (data not shown).

- As for pyramidal cells, the model does not make the distinction between cortico-spinal and cortico-striatal neurons. Connection probabilities highlighted in experimental data oscillate between 10-30% (Kiritani et al., 2012; Kawaguchi et al., 2017). We opted for higher connection probabilities such that excitatory transmission between pyramidal cells (and not only external constant inputs, mimicking for example excitatory inputs from more superficial layers) could also drive their internal dynamics, even with a small number of neurons. This choice is similar to the one made in the model developed by Chadderton et al. (2014).

- The model is probably too simplistic because it treats pyramidal neurons as a pool of point neurons. It is probable that the main effect of STN-DBS is a shift in somato-dendritic axis of

inhibition. For example, in parkinsonian animals, there may be abnormally high levels of excitatory inputs, and the SST neuron activation increases shunting inhibition to block these inputs. These alternatives and shortcoming of the model may be discussed further.

The Referee is absolutely correct that other mechanisms could contribute to the impact of DBS. The purpose of our model was mainly to explore the functions of the asymmetrical connectivity between the two populations of GABAergic interneurons and the pyramidal cells. The model shows that weak or absent feedback excitation to SST neurons is sufficient to account for their higher efficiency at decreasing the firing rate of pyramidal cells under DBS.

We now discuss in more details the fact that alternative features could also underlie or potentially reinforce these effects: in particular, shunting inhibition, but also the fact that synaptic transmission from SST to pyramidal cells presents a short-term facilitation, while PV interneurons have (short-term) depressing synapses onto pyramidal cells (Cardin et al., 2018). This might play a critical role at high-frequency stimulation: indeed, even though both types of interneurons will not be directly driven by 130 Hz, their repeated activations at a lower frequency cycle may still interact with their short-term plasticity dynamics.

Discussion: “Additional mechanisms could further reinforce this effect. The position of SST terminals onto pyramidal dendritic branches, by leading to a local and direct inhibition of excitatory inputs, may explain why SST activation improves information processing compared to PV activation, which exerts a massive shunting (Fig. 4e) through somatic-targeted inhibition^{37,42}. Furthermore, SST-mediated synaptic inhibition depresses only slightly with repeated activation, in contrast to the rapid depression undergone by IPSPs from PV synapses⁴³ and excitatory inputs onto SST cells are also strongly facilitating³⁷. Overall, SST interneurons could thereby be more efficient than PV cells at exerting a sustainable inhibitory influence by filtering excitatory inputs onto pyramidal neurons (rather than simply silencing pyramidal cell output by shunting the inputs), thus resulting in an ameliorated performance of motor functions. A more detailed model, including a multi-compartmental description of pyramidal cells as well as short-term plasticity properties, would allow exploring how these differences in PV and SST physiology could account for the differences in behavioral and processing outcome of their opto-activation.”

Reviewer #3 (Remarks to the Author):

In this manuscript the authors aim to describe the neuronal mechanism through which STN-DBS alleviates motor symptoms in PD with a combination of recording techniques (juxtacellular and intracellular recordings). First, they characterize the changes in activity of motor cortex (M1) neurons in dopamine depleted animals (increase firing rate of pyramidal neurons of 6-OHDA treated rats and mice). Second, they characterize how STN-DBS affects these cortical changes, finding that STN-DBS normalizes the high firing of pyramidal cells in dopamine depleted animals. They hypothesize that local cortical GABAergic network with monosynaptic connection to produce this inhibition. Third, using opto-tagging to identify cortical inhibitory interneurons, they discovered STN DBS decreased the firing rate of PV neurons while it increased firing rate of SST cells. They directly stimulated SST interneurons with optogenetics and find behavioral correction of spontaneous rotations and cross-maze bias.

Last, they describe a neural network model to explore how each cell type responds to STN-DBS and direct stimulation of SST.

These scientific issues are interesting and potentially therapeutically applicable providing a scientific basis for cortical modulation as a therapeutic approach, although how one would modulate specific cell type in human without invasive methods is not clear at this point. The involvement of cortical interneurons in this process is novel and interesting. Network modeling provides interesting insights. However, but there are major gaps to infer causal relationships for their hypothesized mechanisms.

While this manuscript is addressing how STN-DBS is beneficial, the behavioral tests are only carried out with optogenetic stimulation of interneurons. Thus, no comparisons can be made between the efficacy/similarities of STN-DBS and modulating interneuron activity. Additionally, there is an effect of PV opto-stimulation, although it is not as robust (Extended data Fig. 4). Given the lack of improvement in hypoactivity in the open field by SST stimulation, (compare figure 3b and S4c), it would be instructive to know whether STN-DBS in their mouse model has the same effect, and how this compares with therapeutic doses of L-DOPA. Complementary and perhaps more commonly used behavioral paradigms such cylinder test and stepping would have been more convincing as well.

We wish to thank Reviewer#3 for the constructive comments and suggestions, which led us to greatly enrich our study with new experiments, analysis and discussed points.

Indeed, according to Reviewer#3's first comment, we performed these 4 sets of additional experiments:

1) DBS in 6-OHDA-lesioned mice (n=12 mice)

Performing DBS in mice has scarcely been performed, most probably because of the challenge it represents to correctly target the small STN structure with an electrostimulation apparatus light enough for (already weak parkinsonian) mice to handle its weight and not be incapacitated by it. However, Reviewer#3 justifiably underlined the importance of being able to compare the effects of our DBS-inspired optogenetic stimulations to the classical STN-DBS and L-DOPA induced effects on locomotion. We are now presenting these new results in the new Figure 5.

Surprisingly, DBS in mice did not recapitulate the typical motor improvements observed in rats (Darbakay et al., 2003; Maesawa et al., 2004; Gradinaru et al., 2009). Indeed, while locomotor hypoactivity was improved (new Fig. 5b5), none of the asymmetrical locomotor behavior tested (rotations in open field, paw preferences in cylinder test, turn preferences in cross-maze) were alleviated (new Fig. 5b5-d5). These observations are in full accordance with

a very recent study examining the effects of DBS in freely moving mice (Schor and Nelson, 2019); this study is, to our knowledge, the only behavioral investigation concerning DBS in mice.

2) L-Dopa treated 6-OHDA-lesioned mice (n=10 mice)

L-Dopa treatment expectedly improved locomotor activity and reversed all of the asymmetrical locomotor behavior tested (rotations in open field, paw preferences in cylinder test, turn preferences in cross-maze) (new Fig. 5b6-d6).

3) Cylinder test

As recommended by Reviewer#3, we performed new experiments using the cylinder test for all conditions: opto-stimulation in *Sst::ChR2*, *Pv::ChR2* or wild-type mice, DBS in wild-type mice and L-Dopa treated mice (new Fig. 5c1-c6). This new test fully confirmed the observations from the cross-maze: only opto-activation of SST interneurons and L-Dopa treatment conditions reduced the asymmetrical behavior in the cylinder test as well as in the cross-maze.

4) Opto-stimulation in *Sst::ChR2*, *Pv::ChR2* or wild-type mice (additional experiments)

We performed additional experiments (n=+3 opto-PV, n=+2 opto-SST and n=+2 opto-stimulation in wild-type mice), which fully confirmed our previous observations (new Fig. 5).

All these experiments are now illustrated in the new Figures 5 and S4, and in the Results (chapter: Opto-activation of M1 SST interneurons alleviates parkinsonian symptoms in behaving mice; pages 9-11)

Functional manipulations that inhibit SST or PV neurons in combination with DBS would provide a firmer complementary evidence for causal relationship to indicate SST interneurons are necessary as well as sufficient for STN-DBS effect. In addition, it will be helpful to specify whether the model tested that blocking the change in activity of PV or SST neurons abrogates the effect of STN-DBS on the change in pyramidal cell firing rate. Although they note that no decrease in pyramidal cell activity was observed in the absence of DBS-induced current on SST interneurons (Fig.4c), it is not clear what condition this represents.

We fully agree with the reviewer that inhibiting SST (and PV) interneurons during DBS would be an ideal counterpart to our behavioral experiments. Yet two reasons argued against performing these experiments:

First a technical issue: indeed, the timescales for behavioral assessments (several minutes, at least) are not compatible with the continuous silencing of interneurons : inhibitory opsins such as halorhodopsin or archeorhodopsin cannot induce neuronal silence for longer than a few to ~10 seconds since under continuous illumination: (i) the opsin-generated current decreases along the long pulse due to opsin inactivation (Mattis et al., 2011, Nat Meth PMID 22179551), and can be counteracted by intrinsic conductances such as hyperpolarization-activated I_h , which is present in cortical SST interneurons (Ma et al., 2006 JN PMID 16687498), (ii) the risk of thermal damage/aspecific heat-induced activation is increased, (iii) it induces changes in ion homeostasis leading to aspecific side effects that actually counteracts the goal of silencing, such as chloride-potential inversion when using halorhodopsin (Raimondo et al., 2012 Nat Neuro PMID 22729174; Alfonsa et al., 2015 JN PMID 25995461), or neurotransmitter release from the terminals when using archeorhodopsin or chloride-conducting channelrhodopsins (Mahn et al., 2016 Nat Neuro, PMID 26950004). Alternatively, chemogenetic methods, though displaying behavior-compatible timescales,

would likely not silence SST cells completely and would therefore not prevent their recruitment of interneurons by DBS.

Second, a paradigm issue: when we considered alternative methods such a genetically-targeted ablation or constitutive inactivation of an interneuronal population, we opted not to proceed fearing that the complete removal of a class of interneurons would impact cortical physiology so much that the interpretation of the effects of DBS would likely be laborious. This issue in interpreting neuronal silencing results is actually problematic even if transient methods of silencing were available (Wiegert et al., 2017 Neuron PMID 28772120). Ideally, we would like to remove the impact of STN-DBS on the activity of SST or PV interneurons, without affecting their baseline activity.

Nevertheless, the model allowed us to vary or even eliminate the DBS-induced currents received by either PV or SST interneurons (new Fig. 6b). We found that in the absence of DBS-induced currents onto SST interneurons, DBS does not trigger any modulation of pyramidal cell activity. This highlights the crucial importance of SST activation. On the other hand, if no or only a small amount of DBS-induced excitatory current is provided onto PV neurons, SST cells can still efficiently inhibit pyramidal neurons.

Network modeling was used to probe how DBS may affect cortical network activity taking into account stimulation of pyramidal neurons, PV interneurons, and SST interneurons. Based on the parameters selected, the modeling did recapitulate many of the experimental findings, though there were some inconsistencies. The authors use the model to suggest that the decrease in cortical pyramidal cell activity by STN-DBS or SST activation can “preserve” network processing of artificial inputs, which is impaired in the simulated PD condition and suggest that this is the mechanism by which STN-DBS confers its therapeutic benefit. Though these are theoretical studies, the authors should comment on how these findings could be tested.

Overall, there is no insight as to why SST neuronal effect predominates the STN-DBS effect. Is it due to differential connectivity of SST interneurons to STN efferents? The authors note asymmetrical connectivity of SST interneurons, but further elaboration is necessary.

To answer Reviewer#3 comment, we performed new analysis and modeling (new Fig. 6), and detail our answer in the Discussion.

1. Testing the theoretical findings: We now discuss in more detail how the model could be further tested. The theoretical measures we apply to assess changes in information processing capabilities of the network could indeed be adapted to experimental settings. We propose here different ways of testing changes in the information relayed by pyramidal cells, in relation to some previous findings in the parkinsonian condition:

- First and closely related to the theoretical measures we have used, testing changes in neuron selectivity for specific movements or muscle control could be achieved. Such measures could echo previous reports of a loss of segregation of sensory or motor maps in the basal ganglia in the parkinsonian condition, with for example striatal units or GPi neurons showing less spatial or motor specificity (Rothblat and Schneider, 1995 ; Cho et al., 2002 ; Boraud et al., 2000). One could hypothesize that hyperexcitability of pyramidal cells, along with changes in inhibition properties at the cortical level, could in part lead to such downstream effects, with similar changes potentially visible at the cortical level itself. This would involve measuring the properties of neuronal selectivity at the cortical level using extracellular recordings during specific movements.

- Alternatively, measures of entropy and mutual information, along with decoding approaches, could be used in experiments: recording from multiple cortical neurons simultaneously during several trials of a sensorimotor task, or in the presence of different

sensory stimuli, could lead to estimates of the diversity of spiking patterns and of their information content in relation to the task/stimuli. The decoding algorithms could be used on different periods of the task, and may highlight changes in accuracy at specific time points (such as for example during the initiation of movements) related to the parkinsonian motor symptoms. To test more precisely the impact of cortical hyperexcitability on information processing, one could also manipulate during the analyses the recorded spike train properties (for example by randomly removing/adding spikes and analyzing how this may affect the decoding accuracy, or by doing the analysis on normalized firing rate patterns).

This has been added to the Discussion:

" Interestingly, the methodology used for testing the information processing capabilities of the network (Fig. 7) could be applied to experimental recordings: a modulation of the correlation between the execution of specific motor patterns and large-scale in vivo single-unit activities in M1 could be found by comparing sham, 6-OHDA lesioned mice, with or without DBS or SST opto-activation, and changes in the variability of the responses over different trials could be evaluated. In addition, the segregation of sensorimotor maps is degraded in the basal ganglia in the parkinsonian condition^{52,53}. Our model prediction, that DBS or SST opto-activation would restore neuronal selectivity by decreasing cortical hyperactivity, could also be tested by means of decoding algorithms trained on a subset of trials in which different types of movements would be initiated."

2. Connectivity:

In the model, we offer one possible insight as to why SST neuronal effect predominates the STN-DBS effect, that is the asymmetrical connectivity between SST and pyramidal neurons. Several hypotheses, including a differential connectivity of SST interneurons to SST efferents, could also play an important role. Yet, the precise pathways through which SST interneurons are recruited are not really known and are thus not included in the model; we are discussing this point in the revised version of our Discussion.

" Several pathways could account for SST recruitment by DBS: antidromic activation of the cortico-subthalamic fibers, orthodromic activation of subthalamo-cortical fibers, and/or basal ganglia-thalamo-cortical loops^{4,11}. The activation of SST interneurons by antidromic axonal reflex is unlikely to occur since cortico-STN pyramidal cells lack collaterals to SST interneurons²⁴. The orthodromic STN-cortex pathway could recruit SST neurons, since it mainly targets superficial cortical layers³⁴, where pyramidal cells directly connect to L5 SST interneurons³⁵. Concurrently, cortical effects of DBS could occur via recruitment of other basal ganglia-thalamo-cortical loops. Other M1 interneurons than PV and SST might be at play during DBS, such as VIP cells³⁶, since their inhibition would result in the disinhibition of SST cells. The concurrent recruitment of these anatomical pathways likely results in a new cortico-basal-ganglia wide state of equilibrium. This DBS-induced state stabilizes over time, which would explain the observed onset and offset delays in pyramidal and SST responses to DBS."

As for the importance of the asymmetrical connectivity, this configuration enables SST neurons to be more independent from pyramidal cell activity, in contrast to PV neurons, which receive feedback excitation, and are thus more likely to be entrained in the same dynamics as pyramidal cells in the presence of an additional stimulation – such as the excitatory currents modeling DBS stimulation. This is indeed what we observe in both experimental and modeling results : the decrease in pyramidal cells firing rate is accompanied by a decrease in PV interneurons activity.

Furthermore, we now included in the manuscript an additional simulation (Fig. 6c) in which we add and vary the strength of an excitatory connection from pyramidal to SST cells and observe the impact on the firing rates of each population. More precisely, in order to maintain the same excitability of SST neurons, we also decrease the constant external input on SST neurons while increasing the excitatory synaptic strength on SST neurons, such that the firing rates of SST neurons remain the same in the two conditions. The results indicate that adding a feedback excitatory connection onto SST neurons leads to the progressive loss of DBS effect on pyramidal cells, as we increase the strength of this synaptic weight.

Fig. 1 – the authors interpret the pyramidal neuron hyperpolarization caused by DBS to be due to GABAergic inhibition from synaptic input. Could the authors comment on whether a similar hyperpolarization of pyramidal neuron may also be seen if the pyramidal cell is made to fire at a high rate (e.g., 130 Hz) for a prolonged period of stimulation (e.g., as may be expected from DBS antidromic current, or antidromic action potential), which may occur cell-intrinsically via slow afterhyperpolarization via opening of calcium/sodium-gated potassium channels? Unlike a GABAergic mechanism, this would predict the hyperpolarization would still be seen in the presence of AMPA/NMDA/GABA blocker.

Indeed, in antidromically activated pyramidal cells firing at 130 Hz over a prolonged period, there could be a hyperpolarization resulting from cell-intrinsic currents. However, (1) the antidromically activated cells that we recorded were not able to elicit spikes at 130 Hz for more than a few pulses (Fig. S1d), (2) the hyperpolarization along sustained DBS is present whether cells are antidromically activated or not, and (3) the hyperpolarization is maintained while their firing rate decreases and even in the absence of spikes (Fig. S1d).

Lastly, the evoked current (Fig. 2d) reverses at -71 mV, which would not be expected for potassium channels

While we cannot exclude that the observed DBS-induced hyperpolarization of pyramidal cells can involve additional mechanisms on top of GABA transmission, formally testing this would be difficult *in vivo*, where application of GABA blockers would lead to epileptic activity.

The authors mention that “in cells that were antidromically activated by STN stimulation,” (line 55): what % of pyramidal cells showed antidromic activation?

We observed antidromic activation in n=3 pyramidal cells, i.e. in 10% of the intracellularly recorded neurons.

This information has been added in the Results part.

“In a subset of cells (n=3) that were antidromically activated by STN stimulation, the antidromically-evoked action potentials were rapidly shunted and a marked hyperpolarization was observed (Fig. S1d).”

Fig. 2d - The authors should comment on why after DBS stimulation has been turned on, there seems to be a time-lag of 1 time-bin (10 s) for pyramidal neurons, and time-lag of 1-2 time-bins (10-20 s) for Sst neurons, before firing rates changed. And the possible long-lasting effect of DBS for 1-2 time-bins after it's switched off?

These time-lags at the offset of the DBS are indeed very interesting points to comment.

DBS acts on basal ganglia nuclei as well on the cortex through several anatomical pathways, including antidromic activation of cortico-subthalamic fibers, orthodromic activation of subthalamo-cortical fibers, and basal ganglia-thalamo-cortical loops (Deniau et al., 2010; McIntyre et al., 2018). All these pathways involve different relays, and very different delays, from direct antidromic activation of pyramidal neuron to multi-synaptic trans-basalo-thalamo-

cortical loops. Moreover, the relays are themselves interconnected, so the "longer" pathways start to interact with the "faster" pathways after the onset of DBS. Therefore, it is likely that the global "sum" of these interacting effects on the cortex reaches an equilibrium only after a transient phase (and similarly, a transient phase is seen at the offset).

This point is now added in the Discussion (p17):

“Several pathways could account for SST recruitment by DBS: antidromic activation of the cortico-subthalamic fibers, orthodromic activation of subthalamo-cortical fibers, and/or basal ganglia-thalamo-cortical loops^{4,11}. The activation of SST interneurons by antidromic axonal reflex is unlikely to occur since cortico-STN pyramidal cells lack collaterals to SST interneurons²⁴. The orthodromic STN-cortex pathway could recruit SST neurons, since it mainly targets superficial cortical layers³⁴, where pyramidal cells directly connect to L5 SST interneurons³⁵. Concurrently, cortical effects of DBS could occur via recruitment of other basal ganglia-thalamo-cortical loops. Other MI interneurons than PV and SST might be at play during DBS, such as VIP cells³⁶, since their inhibition would result in the disinhibition of SST cells. The concurrent recruitment of these anatomical pathways likely results in a new cortico-basal-ganglia wide state of equilibrium. This DBS-induced state stabilizes over time, which would explain the observed onset and offset delays in pyramidal and SST responses to DBS.”

Is the same time-lags observed in their simulation model?

Even though, the simulations gave time-lags at the DBS offset, the model does not reproduce exactly the same time-lags as those observed experimentally. As can be seen from the time course of the firing rate of pyramidal cells at the transition DBS-On/DBS-Off, the time-lags remain in a small range of 15-20 ms and can still be observed in the absence of delay in SST (and the second half of pyramidal cells) activation relative to the other half of pyramidal cells (new Fig. 6a). This rapid response is likely due to the fact that DBS, in the model, acts as an instantaneous current to the cells, and to the fact that the model does not take into consideration the multi-synaptic pathways that may be involved in the settlement of the cortical network in a new activity state.

Fig. 2e – criterion for classifying as excited vs inhibited vs non-modulated vs silent is not specified.

This is indeed an important information to add. Neuronal responses to STN-DBS were classified as non-modulated if they did not display significant variation from their spontaneous activity (<5%), and as silent if they displayed an absence of discharge during the course of the recording.

To clarify this point, we added in the Methods section (p31) the following sentence:

“Neuronal responses to DBS were classified as non-modulated if the variation of their spontaneous activity was <5% compared to baseline activity.”

The authors should explain the choice in stimulation parameters for opto-tagging. 100ms is a long time (and it is clear that the PV and SST neurons don't maintain an increased firing rate during the stimulation duration) as well as the intensity (reported as 10mW). These parameters can have long lasting effects on the cells, even though the authors report that they waited for the neurons to “recover”.

We do agree with Reviewer#3 that we cannot be entirely certain that the photo-identification of neuronal populations does not have a significant effect on their spontaneous activity, just like anesthesia could have an effect on spontaneous activity in the long term. Our study focuses on the effect of DBS, and as can be observed in the Figure 3, neuronal activity returns

to baseline activity after the DBS is turned off. This fact advocates for equivalent results whether opto-tagging has an effect on spontaneous activity or not.

For opto-tagging, we used relatively high intensity to make sure the light would reach the deeper layers of the cortex which was our target. Indeed, in these experiments, the optic fiber was placed over the brain surface, to avoid lesioning the cortical network that we were recording from. Even in our high intensity conditions (10 mW at the tip of a 200 μ m-core, 0.5NA fiber), and according to the attenuation model provided in the Optogenetic Resource Center (web.stanford.edu/dlab/cgi-bin/graph/chart.php), the irradiance below \sim 1mm would be lower than the ChR2-H134R activation threshold (0.98 mW/mm², Lin 2011). Decreasing the intensity any further would have risked increasing dramatically the risk of false negative neurons (ChR2-expressing, but not reached by a strong enough light to be activated). Moreover, even with 10 mW, it is likely that the soma of a non-negligible proportion of layer 5 neurons was around or below the depth limit for ChR2-activation threshold; therefore, their opto-response would rely on the activation of the superficial part of their dendritic tree. For these neurons, longer light pulses help maximize the probability of getting a spike in response to opto-activation, by increasing the probability of ChR2 activation near-threshold, and allowing longer depolarization times for dendritic integration. Therefore, we used a long duration pulse in order to be certain that we would not misclassify long-latency responding neurons as non-responding (and indeed, a few SST neurons displayed around and over 10 ms latency; new Fig.S3c2).

To prevent these light parameters to have long-lasting effects on the opto-tagged cells, we used a low frequency for opto-tagging (0.5 Hz). At this rate 100ms pulses represent 5% of duty cycle, which is much smaller than typically used when using ChR2 to activate neuronal populations. The resulting successive depolarization of 100 ms at 0.5 Hz is no different, if not less intense, than common I-V protocols used in vitro or in intracellular recordings to characterize cell populations. Moreover, opto-tagging involved typically only 10-15 pulses, i.e. 20-30 sec, after which neurons were indeed allowed to recover for at least 1 minute and until a stable discharge activity was observed (stabilization of discharge activity usually occurs after 10-30 seconds).

Explanation as to why the 67Hz was chosen for optogenetic stimulation for behavioral assays should be provided. It will be important to note why the “typical” clinical frequency was not used or provide a discussion as to why other frequencies were used. Along the same lines, can the M1 interneurons in each of these experiment actually follow the stimulation frequency applied?

We chose the 67 Hz frequency knowingly, as we did not want all opto-stimulated interneurons in the vicinity of the optic fiber to be activated synchronously at each light pulse. Indeed, since synchronous oscillations are a hallmark of Parkinson’s disease (Singh, 2018, PMID 29381817), we wanted to avoid imposing a strong rhythm to the whole cortical networks. Therefore, we chose a frequency higher than 50 Hz (20 ms period), which PV and SST interneurons would be able to follow. Rather, our goal was that each pulse elicits the activity of a proportion of neurons stochastically picked among the population. We also wanted to avoid choosing a frequency so high that ChR2-H134R would not have time to close, which could lead to the silencing of nearly all neurons with the continuous application of the protocol. We therefore were capped by the tau-off of ChR2: \sim 15-20 ms (Lin et al., 2009 PMID 19254539); we chose the lowest value of this range, which corresponds to 67 Hz (15 ms period).

To validate this choice (67 Hz), we recorded opto-identified PV and SST interneurons during 1-minute continuous stimulation at 13 Hz (a frequency that ChR2-H134R should be able to follow easily), 67 Hz (our chosen frequency) and 130 Hz (the electrical DBS frequency) (new

Supplementary Figure 4a and 4b). Among neurons that were well entrained by 3ms pulses, we observed that at 13 Hz, all PV and SST neurons were able to fire at every light pulse. At 130 Hz, the proportion of responding cells at a given pulse decreased to 7% of the SST and 10% of the PV interneuron (measured during the last 10s of 130 Hz-stimulation), with 45% of the SST and 18% of the PV being nearly or totally silenced (0-1 % success rate per pulse). Opto-stimulating at 67 Hz was able to maintain a mean of 26% of SST and 55% of PV cells responding at a given light pulse during the last 10s of opto-stimulation. Furthermore, the responding cells were not the same at each light pulse, with 82% of the SST and 100% of the PV interneurons participating with a success rate per pulse > 1%. Therefore, stimulating at 67 Hz appeared as a good compromise, allowing a neither negligible (130 Hz) nor overwhelming (13 Hz) proportion of interneurons to fire.

To answer this point, new analysis and additional experiments have been performed, and are now illustrated in the new Supplementary Figure 4a and 4b, and detailed in the Results and Methods sections.

“We chose a stimulation frequency (67Hz) that was close to the opsin tau-off (~15-20ms⁶⁴). Indeed, higher frequencies such as 130Hz (the same frequency as electrical DBS) would lose efficiency quickly, while lower frequencies such as 13 Hz, able to entrain all the neurons at each pulse (Fig. S4a and S4b), might impose pathological-like synchronous activation⁶⁵. DBS stimulation: electrical current (60μs, 120μA, 130Hz).”

For the cross-maze experiments: were the two sessions described (stim, no stim) balanced as to which one went first/second? There is a concern for learning with the increased number of trials. Additionally, these sessions could have been combined and have trials stim/no stim randomly until the 30 turn criteria is met.

For all the mice, the control session of the cross-maze was performed first and the opto-stimulation session second (at least 1 day after). This was the case also in the electrical DBS experiments and levodopa treatment experiments now added to the study (new Fig. 5). This choice was based on the description of this test in Maurice et al. (Cell Reports 2015), and also motivated by the fact that we did not know whether the opto-stimulation might have a post-effect which might induce a bias in combined or balanced sessions: for example, with levodopa treatment, our observations (not included in this study) is that if the levodopa session is done before the saline session, there is a significant residual effect even 3 days later. We are confident that this experimental design does not lead to a bias due to the increased number of trials since:

(i) there is no reward or punishment of any kind in this test, that might induce a learning of a preferred choice. It is free exploration and “trials” are continuous: a new “trial” begins once the mouse has fully entered an arm, and this “chosen arm” from the previous trial becomes the “start arm” of the new trial. Therefore there is no reward-based learning to be expected, nor “punishment-based” (e.g. if the experimenter picked up the mouse after each choice which might cause a stress leading to a decrease in ipsi-choices). The only potential learning that might occur could be the motor habit of making ipsilateral choices, but the mouse already expresses this behavior continuously, because of the lesion, in the absence of test (their spontaneous exploration in the homecage is always towards ipsilateral direction). Lastly, even if there was a developing motor habit of ipsilateral choices specific to the cross-maze, this would rather tend to decrease our chances to see an effect of opto-stimulation (or other treatment).

(ii) To compensate for the unbalanced design, we had also recorded another control session in the opto-stimulated and DBS-stimulated mice, after the stimulation session (at least 1 day after). If a bias due to the repetition of sessions existed, it would have been enhanced with this

extra-session, which was not observed. This new data has now been included in the study (new Fig. 5d1-d6).

(iii) More importantly, the hypothesis of a bias can be excluded by the results of the control group, wild-type non-opsin expressing mice, which underwent the exact same experimental design, and did not display any significant change across the repeated sessions (new Fig. 5d4).

The data corresponding to the extra control session, and a clarification of the cross-maze protocol have been included in the new Figure 5 and in the Methods section, respectively.

It would be helpful to the reader to explain in the figures which animals (rats vs mice) were used in the experiments presented (ie with the experimental set-up image).

This is now clearly indicated in each figure and the corresponding title of the legends which animals were used. As suggested by Reviewer#3, we are now clearly indicating in each experimental set-up image which rodents were used.

It is unclear what cells are included in the “cortical cells” that decreased firing rate with STN stimulation (line 47). Is it just pyramidal cells or all cells?

It was indeed a mistake, and we replaced “cortical cells” by “pyramidal cells”.

In the revised version of our manuscript, we added a new panel (panel a) in the Supplementary Figure 1 (see figure below) showing how pyramidal cells were identified based on their waveforms.

Figure legend: Representative trace of a single action potential of a pyramidal neuron recorded in the rat orofacial motor cortex. Left panel: dashed lines delineate waveform width, from start to negative trough. Plot indicates mean spontaneous firing rate against waveform duration for all recorded neurons in Sham and 6-OHDA rats. Neurons with broad waveforms (> 1 ms) were considered as putative pyramidal neurons, the rest were excluded from analysis.

Fig. 4c – why is DBS of PV (vs. Sst) less effective in reducing pyramidal firing here, but seems to be more effective in Supplementary Fig.5C?

The conditions between Fig. 4c (new Fig 6b) and Fig. S5c are distinct and both convey the fact that PV neurons are relatively less effective than SST cells in reducing pyramidal firing.

Indeed, in previous Fig 4c, only half of pyramidal cells receives a fixed amount of current (120 pA at 130Hz), while the other half does not receive any current. In the new Fig. S5c, SST and PV neuron activation is varied, with only one half of pyramidal cells receiving DBS-induced excitatory current.

In the new Fig 6b (equivalent to the previous Fig. 4c) we vary the amount of current sent to PV and SST neurons, and find that PV neurons alone can hardly change the firing rate of

pyramidal cells (if both receive positive excitatory currents), as visible in the bottom right square of the heatmap.

In the new Fig. S5c (right), we vary the amount of current to PV and to half of pyramidal neurons. But in this case, both the other half of pyramidal cells and all SST neurons receive a fixed amount of current (120 pA at 130 Hz). In the new Fig. S5c (left), we vary the amount of current to SST and to half of pyramidal neurons. Again, both the other half of pyramidal cells and all PV neurons receive a fixed amount of current (120 pA at 130 Hz). Thus, the comparable condition of new Fig. 6b bottom right square is in fact in Fig. S5c (left) at the bottom horizontal line, in the middle of the heatmap when SST neurons and the other half of pyramidal cells receives no current: we can observe again that PV neurons alone hardly change the firing rate of pyramidal neurons.

We clarified in the new Figures 6 and S5 the different conditions used to construct these heatmaps.

Supplementary Fig. 6a – why does the simulation model predict that opto-stimulation of Sst leads to a high asymptotic Sst firing rate (~40 Hz) while opto-stimulation of PV leads to lower asymptotic PV firing rate (~20 Hz), when the opposite seem true with real experimental data shown in Supplementary Fig. 7a? Does this have downstream consequences for the simulation model? (E.g., maybe having PV firing to much in simulation causes pyramidal cells to be too inhibited? If yes, how to reconcile?)

The referee is right to pinpoint this difference between the modeling results and the experimental data, even though the resulting discharge rates of pyramidal cells in the model remained very close to the experimental data. In the present model, two main reasons underlie why SST neurons have higher firing rates compared to PV neurons during the high frequency pulses of opto-stimulation:

- the strong self-inhibitory coupling between PV neurons prevents them from getting too excited and reaching high firing rates
- the relatively small value for the spike-triggered adaptation for SST neurons enable them to keep spiking at a relatively higher frequency compared to the experimental results.

To reduce this discrepancy, we have decided to modify some of the intrinsic properties of PV and SST neurons in order to get closer to those experimental results (see new TableS1); all simulations were redone accordingly (new Fig. 6, 7, S5, S6 and S7). We thus increased the spike-triggered adaption for SST neurons (from $b=40$ to $b=90$, remaining in the range of values used by Litwin-Kumar et al., 2016 and Brette and Gerstner, 2005). In order to make PV neurons more responsive, we also hyperpolarized their spiking threshold to $V_{\text{threshold}} = -52$ mV. To maintain the difference between PV and SST interneurons (SST spiking threshold is lower compared to PV neurons), we decreased SST threshold to $V_{\text{threshold}} = -53$ mV, while reducing their constant external input from $I_{\text{ext}} = 35$ to $I_{\text{ext}} = 25$ pA.

We chose not to change the synaptic connectivity rules since PV strong self-inhibition appears to be a widespread and well-described feature in the literature.

We also decided to focus in the main figures on the impact of a more intense opto-activation protocol, with a 600 pA current injected at 67 Hz to either PV or SST neurons (in comparison to 400 pA in the previous version), thus leading to higher firing rates for PV neurons up to 40-50 Hz.

These modifications of the network architecture did not lead to any significant changes in the impact of DBS on pyramidal cells firing and in the improvement of information processing

capacities under DBS and SST-opto-activation. This reinforces the robustness of the model, tested over a wide variety of parameters (data not shown) and the message it conveys.

In reference to the second question of the Reviewer#3, opto-stimulation of PV interneurons indeed leads to a stronger silencing of pyramidal cell activity compared to SST opto-stimulation; and even stronger with the new set of parameters, as we have increased PV firing rates. This effect is mediated by the stronger unity synaptic weights of PV to pyramidal cells connections included in our model (Tanaka et al., 2011; Pfeffer et al., 2013).

Yet, the level of sparsity is not the only factor determining the information processing capacities of the network, as described in the new Supplementary Figure 7. Indeed, in this figure, we have added the results of simulations testing the impact of various opto-stimulation current intensities on network activity in response to stimuli. Importantly, even when the firing rate of pyramidal cells were identical under both PV or SST opto-activation (or when the firing rate of pyramidal cells is lower under high-intensity stimulation of SST compared to low-intensity stimulation of PV neurons), the correlation coefficients and classifier accuracy under SST opto-activation remain higher compared to the PV opto-activation condition. Thus, these results highlight the importance of the temporal patterning of pyramidal cells activity in response to a stimulus.

Yet, of course, when an excessive silencing of pyramidal cells activity is induced by very high opto-activation current intensities (typically at 800 pA or more), pyramidal cells fail to trigger action potentials in response to some stimuli – this was especially visible under PV stimulation – thus necessarily degrading stimulus encoding and its recognition by downstream structures.

This discussion has been added in the results of the computational part: *“Importantly, the differential impact of PV and SST opto-activation remained consistent over a large range of current intensities. The reduced efficacy of PV opto-activation was not only due to the higher silencing level of pyramidal cells but also to the induction of oscillatory-like responses (Fig. S7a-b).”*

All simulations have been redone with the new parameters (see new Table S1), and the results are presented in the new Figures 6, 7, S5, S6 and S7.

It will be helpful to note how the coordinates for STN was chosen. This structure has a particular organization also receiving input from the PFC. Similarly, how were the coordinates for M1 chosen? The location of all recordings and stimulation seems to be of the anterior portion.

The approximate location of the STN was first chosen based on the Franklin & Paxinos mouse brain atlas. In addition, its precise location was chosen based on electrophysiological identification of STN neuronal discharges, as described in the Supplementary Figure 3. We attempted to target the lateral STN, as it receives inputs from the motor cortex. Concerning the coordinates for the M1 cortex, they were also chosen based on the Franklin & Paxinos mouse brain atlas. However, the cluttering of our setup (i.e. the sizes of the holders for the stimulation electrode in the STN and for the optical fiber for the M1) prohibited us from exploring the more posterior part of the M1.

Please correct/align the labels in Figure 2 correctly (bottom of panel b and c).
This has been corrected.

Line 85: should say juxtacellularly labeled.
This has been corrected.

The exclusion criteria was defined at “spontaneous ipsilateral rotations”. How long was the testing session? Typically the number of amphetamine rotations per min are reported and an a priori minimum is set.

In the present study, we did not use amphetamine injections to induce rotations, neither for the exclusion criteria prior to behavioral testing, nor for open field rotation and locomotor activity test. The exclusion criteria was tested during 6-OHDA lesion recovery and prior to the inclusion of the mice in the behavioral testing, by daily observations for a few minutes in their homecage and on the scale used for weighing. In our previous experience of the MFB unilateral lesion model in mice, mice that fail to show spontaneous ipsilateral rotations in the homecage and when placed on the weighing scale (i.e., explore indifferently towards ipsi- and contra-lateral sides), together with a lack of weight loss, display little to no lesion in TH immunohistochemistry *a posteriori*. We therefore excluded these mice from the current study, before even performing the second surgery session (optic fiber or electrode implantation).

According to Reviewer#3 comment, the exclusion criteria has been clarified in the Methods section (p29).

« Mice who failed to display spontaneous ipsilateral rotations (i.e., were seen to explore indifferently towards ipsi- and contralateral sides when observed in their homecage and on the scale during daily weighing, without amphetamine challenge) and weight loss in post-lesion recovery were excluded from the study.»

What do the authors consider an opto-response latency? This should be reported in the methods.

The opto-response latency is calculated as the time elapsed between the onset of the light pulse and the occurrence of the first spike within the light pulse.

As suggested by Reviewer#3, this information has been added to the Methods part (p31).

« the opto-response latency was the time elapsed between the onset of the light pulse and the first spike within the light pulse (mean, SD and CV of the latency were calculated). »

Though the authors state in the statistics methods that unless otherwise reported results are displayed as mean±SEM, it would be helpful to the reader to have it stated in the individual figures.

As suggested by Reviewer#3, this information has been added in every relevant legend.

Reviewers' Comments:

Reviewer #1:

Remarks to the Author:

Authors provided a complete and very detailed revision of the paper, including several new experiments and analysis. All my issues have been clarified and the paper has largely improved. I have no more request.

Reviewer #2:

Remarks to the Author:

The authors' responses were very detailed and addressed all of my comments. I do not have anything else to add.

I remain enthusiastic about the manuscript, based on the same reasons as I listed in the first round of review: The writing is crisp and clear. The data are high-quality, and rigorously collected using contemporary electrophysiological, optogenetic and behavioral techniques in rodents. The analyses are sound. This is an excellent study.

Reviewer #3:

Remarks to the Author:

This manuscript has been extensively revised, adding new experiments including behavioral outcomes, more clear data presentation, and expansion of discussion.

Notably, the authors show new data that the behavioral improvement afforded by DBS vs. Sst-opto stimulation in mice differs: DBS improves akinesia and not asymmetry, whereas SST-opto stimulation improves asymmetry but not akinesia. Although the discussion touches on the idea that these differential effects may arise through modulation of additional circuits (i.e., basal ganglia) by STN-DBS, differences in the pattern and extent of the increase in Sst neuron activity differ in each case, making it difficult to compare the two. For instance, Sst neuron firing increases gradually, peaking at the end of the DBS stimulation period and increase to a median of ~ 1 Hz, whereas with optogenetic stimulation of Sst cells, the firing rate peak near the onset of light and decreases over the duration of stimulation and reaches about 25 Hz. It could be that DBS and Sst stimulation are separate phenomenon in this model. In addition, the effect of levodopa with the chosen dose of 6 mg/kg is quite strong and overcompensates the deficit whereas DBS parameters have been chosen to be below dyskinesia threshold, which makes the comparison of the effects somewhat limited without "dose" curves of each treatment. SST activation may be simpler closer to the levodopa dose rather than the gentle DBS parameters. Differences in behavioral outcomes may reflect this issue. Therefore, it is difficult to conclude that "From our results, the main beneficial effects of DBS seem to be conveyed by the activation of SST cells" in line 380 without significant qualification.

Minor points:

1) Abstract: "A computational model highlights that the decrease in pyramidal neuron activity can restore information processing capabilities."

- I'd be more careful with this sentence because Fig 6d shows that, according to the model, opto-PV strongly decreased pyramidal neuron activity, but at the same time was quite bad at restoring information according to Fig 7c-d.

2) Discussion, P. 20, Lines 413 to 491: "Furthermore, SST-mediated synaptic inhibition depresses only slightly with repeated activation, in contrast to the rapid depression undergone by IPSPs from PV synapse..."

- This sentence seems to contradict Fig. 4e?

3) Figure 7b: Parkinson and Parkinson + DBS plots seem to be exactly the same figure

4) Vehicle is misspelled in Figure 2a

REVIEWERS' COMMENTS:

Reviewer #1 (Remarks to the Author):

Authors provided a complete and very detailed revision of the paper, including several new experiments and analysis. All my issues have been clarified and the paper has largely improved. I have no more request.

Reviewer #2 (Remarks to the Author):

The authors' responses were very detailed and addressed all of my comments. I do not have anything else to add.

I remain enthusiastic about the manuscript, based on the same reasons as I listed in the first round of review: The writing is crisp and clear. The data are high-quality, and rigorously collected using contemporary electrophysiological, optogenetic and behavioral techniques in rodents. The analyses are sound. This is an excellent study.

Reviewer #3 (Remarks to the Author):

This manuscript has been extensively revised, adding new experiments including behavioral outcomes, more clear data presentation, and expansion of discussion. Notably, the authors show new data that the behavioral improvement afforded by DBS vs. Sst-opto stimulation in mice differs: DBS improves akinesia and not asymmetry, whereas SST-opto stimulation improves asymmetry but not akinesia. Although the discussion touches on the idea that these differential effects may arise through modulation of additional circuits (i.e., basal ganglia) by STN-DBS, differences in the pattern and extent of the increase in Sst neuron activity differ in each case, making it difficult to compare the two. For instance, Sst neuron firing increases gradually, peaking at the end of the DBS stimulation period and increase to a median of ~1 Hz, whereas with optogenetic stimulation of Sst cells, the firing rate peak near the onset of light and decreases over the duration of stimulation and reaches about 25 Hz. It could be that DBS and Sst stimulation are separate phenomenon in this model. In addition, the effect of levodopa with the chosen dose of 6 mg/kg is quite strong and overcompensates the deficit whereas DBS parameters have been chosen to be below dyskinesia threshold, which makes the comparison of the effects somewhat limited without “dose” curves of each treatment. SST activation may be simpler closer to the levodopa dose rather than the gentle DBS parameters. Differences in behavioral outcomes may reflect this issue. Therefore, it is difficult to conclude that “From our results, the main beneficial effects of DBS seem to be conveyed by the activation of SST cells” in line 380 without significant qualification.

We agree with Reviewer #3 the difficulty to compare the effects of a dose of systemic pharmacological treatment (levodopa), to those of an intensity of electrical deep brain stimulation (DBS), and those of a single protocol of optical stimulation in the motor cortex (M1). This is the reason why we chose not include such comparison in the first version of the article, and performed these experiments it because of reviewer suggestion.

Schor and Nelson (2019) built a dose-response of DBS parameters in mice (up to 120 μ s/300 μ A pulses, which can hardly be considered “gentle” since they evoke reliable dyskinesia in half the mice), and report an improved velocity (and higher probability of dyskinesia) with increasing intensity, frequency or pulse length, but a lack of effect on rotational bias. Therefore, the lack of effect on asymmetry in our case is unlikely to be due to too gentle stimulation parameters for DBS.

We chose the dose of 6 mg/kg of L-dopa based on Lundblad et al. (2004): in their study it is the lowest dose that relieves forelimb akinesia and ipsilateral rotations in MFB-lesioned (or striatum-lesioned) 6-OHDA mice. They do see the same reversion of effect as we do in open field rotations (strong contralateral rotation behavior at 6mg/kg), whereas at 2 mg/kg the mice are still significantly turning towards the ipsilateral side. Their forelimb akinesia results similarly show no significant effect at 2 mg/kg, and a relief, but no reversion at 6mg/kg. Therefore, it is likely that the range of L-dopa dose for which a relief would be observed without a reversion is quite limited, and variable depending on other experimental conditions. As mentioned in our Discussion, this may be due to the unilateral 6-OHDA model, with compensations in the dopamine receptor transduction efficiency happening during the post-operative recovery in the dopamine-deprived lesioned hemisphere: once this imbalance between hemispheres is set, a systemic treatment such as an IP injection is likely to induce reversions, as soon as the dose is sufficient to be effective on symptoms. Therefore, while we cannot exclude it formally, it appears that the choice of parameters is unlikely the key reason for the observed differences.

We agree with Reviewer#3 that SST activity under DBS vs optical stimulation is different (and unknown in the case of L-dopa treatment). But even with dose-response curves and electrophysiological data for each strategy (allowing to set each parameter depending on a common effect on SST neurons), it would be difficult to do a point-to-point comparison of their effects since the spatial extent of their direct effects is widely different: whole brain for L-dopa, a potentially large part of frontal cortex (either directly because of the highly convergent input from large areas of frontal cortex to the small-sized STN, or indirectly through the divergent output of basal ganglia on frontal thalamo-cortical networks) as well as direct effect on SNr and GPe through electrical DBS, and at maximum a cubic millimeter of M1 with single-fiber optical stimulation. We agree with Reviewer#3 that these reasons (differences in temporal, spatial extent and magnitude of effect on cortex, and SST interneurons in particular, as well as direct effects on other structures for DBS and L-dopa), more than the precise parameters used in our study, drive the different outcome of the three strategies, and that is what we tried to express in our Discussion.

We now added the following sentence in the Discussion to better convey this notion:

“Alternatively, the magnitude or spatial extent of DBS cortical effects (milder stimulation of SST neurons reaching potentially larger regions of frontal cortex than single fiber opto-stimulation).”

The statement that “the main beneficial effects of DBS seem to be conveyed by the activation of SST cells” was not meant to rely on the comparison of the behavioral results, but on (1) the electrophysiological data showing that DBS concurrently inhibits pyramidal cells and PV cells while increasing SST firing rate (in a highly correlated time course with pyramidal cells), and (2) the modeling results showing that the DBS-induced current on SST population was necessary for DBS to induce a beneficial effect on cortical information processing (while DBS-induced current in PV was not). Although these results strongly point to a role of SST activation in DBS-induced normalization of pyramidal cell activity, they are not a direct proof, and the step from cortical electrophysiological and computational beneficial results to behaviorally beneficial results is not straightforward. Therefore, we modified the Discussion sentence accordingly to clarify these points:

“The correlated time courses of the decreased hyperactivity of pyramidal cell and SST activation upon DBS combined with our modeling result suggest that the main beneficial effects of DBS on normalization of cortical activity and processing capabilities seem to be conveyed by the activation of SST cells.”

Minor points:

1) Abstract: “A computational model highlights that the decrease in pyramidal neuron activity can restore information processing capabilities.” I’d be more careful with this sentence because Fig 6d shows that, according to the model, opto-PV strongly decreased pyramidal neuron activity, but at the same time was quite bad at restoring information according to Fig 7c-d.

According to Reviewer#3 comment, we have modified this sentence for “A computational model highlights that a decrease in pyramidal neuron activity induced by DBS or by a stimulation of cortical somatostatin interneurons can restore information processing capabilities.”

2) Discussion, P. 20, Lines 413 to 491: “Furthermore, SST-mediated synaptic inhibition depresses only slightly with repeated activation, in contrast to the rapid depression undergone by IPSPs from PV synapse...” This sentence seems to contradict Fig. 4e?

We do agree with reviewer#3 that this sentence can at first appear to contradict our data. However, in our *in vivo* patch-clamp experiments, we observed the combined result of the opto-activation of PV or SST interneurons (@ 67Hz) and the short-term plasticity of interneuron -> pyramidal synapses. In contrast, the review cited in the discussion is based on double patch-clamp studies of unitary IPSCs (such as Hofer et al., Nat Neuro 2011; Ma et al., J Neurosci 2012), which excludes the differences in opto-stimulation efficiency (stronger efficiency in PV than SST, see Fig. S4a). Moreover, these studies measured short-term plasticity at 20-25 Hz, and short-term plasticity strongly depend on the stimulation frequency. Considering that the activation of SST interneurons is milder when elicited by electrical DBS compared to cortical opto-activation, our results in Fig.4e are more suited to interpret the differential effect of PV vs SST opto-activation, while the literature cited in the Discussion pertains more to the reason why SST neurons constitute a better relay of DBS effect on pyramidal neurons than PV cells. Still, we agree with Reviewer#3 that this nuance is not clear in the discussion. Therefore, to avoid confusion (and because of limited length of the manuscript) we removed this sentence in this revised version.

3) Figure 7b: Parkinson and Parkinson + DBS plots seem to be exactly the same figure
Our mistake: it was indeed twice the same graph (Parkinson condition). This has been corrected, and the proper “Parkinson+DBS” graph is now in Fig. 7b.

4) Vehicle is misspelled in Figure 2a
This has been corrected.